# A facet atlas: Visualizing networks that describe the blends, cores, and peripheries of personality structure

**Ted Schwaba***, **Mijke Rhemtulla, Christopher J. Hopwood, Wiebke Bleidorn**

Department of Psychology, University of California, Davis, Davis, California, United States of America

* tedschwaba@gmail.com

**Data Availability Statement:** The raw ESCS data are publicly available at https://dataverse.harvard.edu/dataverse/ESCS-Data, and the cleaned data

## Abstract

We created a facet atlas that maps the interrelations between facet scales from 13 hierarchical personality inventories to provide a practically useful, transtheoretical description of lower-level personality traits. We generated this atlas by estimating a series of network models that visualize the correlations among 268 facet scales administered to the Eugene-Springfield Community Sample (Ns = 571–948). As expected, most facets contained a blend of content from multiple Big Five domains and were part of multiple Big Five networks. We identified core and peripheral facets for each Big Five domain. Results from this study resolve some inconsistencies in facet placement across instruments and highlight the complexity of personality structure relative to the constraints of traditional hierarchical models that impose simple structure. This facet atlas (also available as an online point-and-click app at tedschwaba.shinyapps.io/appdata/) provides a guide for researchers who wish to measure a domain with a limited set of facets as well as information about the core and periphery of each personality domain. To illustrate the value of a facet atlas in applied and theoretical settings, we examined the network structure of scales measuring impulsivity and tested structural hypotheses from the Big Five Aspect Scales inventory.

## Introduction

Over the past few decades, a general consensus has emerged regarding the structure of individual differences in higher order personality traits [1]. In this hierarchical model, two superordinate factors (alpha and beta) subsume five broad domains (the Big Five; extraversion, agreeableness, conscientiousness, neuroticism, and openness to experience), which can be subdivided into narrower facets and even narrower nuances [2–4]. Similar hierarchical models have been developed for personality pathology [5,6] and mental disorders more generally [7].

Although the field has come to a general agreement concerning the number and content of factors at the higher levels of the hierarchy (but see [8]), there is very little consensus regarding the lower-level structure of facets [9–11] (for the purposes of this paper, we define a personality facet as any personality trait that is narrower than a domain yet broader than a specific behavioral nuance). Common lower-order measures range from as few as 10 lower-order facets [12]

and R scripts used in this study are available at
https://osf.io/cjz8e

**Funding:** The author(s) received no specific funding for this work.

**Competing interests:** The authors have declared that no competing interests exist.

to as many as 45 [13]. Some of these measures were developed specifically to measure the spaces below each of the Big Five domains (e.g. [10]), whereas other measures subdivide the facet space in unique ways that do not correspond closely to the Big Five (e.g. [14, 15]). There is also little agreement about how to name these facets. This leaves room for jingle fallacies in which facet scales with similar labels measure different constructs, as well as jangle fallacies, in which differently-labeled facets measure the same construct [16]. Together, these issues have complicated the development of both personality tests and theory [17].

Faced with similar problems, the field of genetics has created atlases that allow researchers to easily identify a particular gene's chromosomal region, function, and co-occurrence with other genes (e.g., [18]). These atlases help researchers understand individual genes, facilitate communication between scholars, and improve the coherence of research programs. Atlases have similarly been used in clinical personality assessment to standardize interpretation of test scores across clinicians (e.g., [19]).

We believe the idea of an atlas can be fruitfully applied to the study of personality facets. A facet atlas could be used as a practical reference guide for researchers or clinicians with personality data or as an investigative resource for researchers exploring questions regarding facet structure. We accordingly created a facet atlas that summarizes the interrelationships between existing facet scales and the associations between facets and the Big Five domains. To do this, we estimated a series of network models that visualize and summarize the patterns of correlations between 268 facet scales from 13 hierarchical personality measures administered to the Eugene-Springfield Community Sample (ESCS; see [20]). We addressed three questions to demonstrate the value of this facet atlas for personality assessment and theory. First, to what extent are facets *blended* representations of multiple Big Five domains? Second, which facets compose the *core and periphery* of each Big Five domain? Third, how can this facet atlas be used to better understand *particular constructs and instruments*?

## Blended facets

Personality traits do not have a simple structure [21,22]. Even measures designed to maximize each facet's correspondence with a single domain often contain facets that have substantial associations with multiple domains [21]. For example, although interpersonal warmth is classified as a facet of extraversion in some measures and as a facet of agreeableness in others [10], it is a blend of both agreeableness and extraversion [23]. As such, the placement of warmth within one or the other of these domains is somewhat arbitrary.

Some domains form a circumplex, where common variance between the two is occupied by meaningful traits [13]. This has been well-documented for extraversion (agency) and agreeableness (communion), the domains that organize the interpersonal circumplex [23–25]. By capturing all blends of these two domains, this model has allowed interpersonal researchers to identify evidenced-based principles for how dyads interact [26,27] and has provided interpersonal clinicians with a useful rubric for case formulations about individual patients [28,29].

One past effort sought to create a circumplicial periodic table of blended facets, using 8 questionnaires assessed in the same dataset as the present study and two questionnaires assessed in a different sample [30]. Woods and Anderson (2016) identified 22 common blends and provided a term for each (for example, high conscientiousness and low extraversion was termed cautiousness). Notably, although they found content to represent most Big Five blends, they found very little content measuring either positive blends or contrasts of agreeableness and conscientiousness.

In the present study, we investigate if this finding holds when additional facets are included from other hierarchical personality inventories, and we investigate whether some facets

contain blends of three or more Big Five domains. Indeed, some areas between higher-order domains that are currently not well measured. These spaces may be unmeasured because of test development biases favoring simple structure, and improving coverage of these spaces might enhance the comprehensiveness of personality trait measurement. Alternatively, these uncommon blends might reflect necessarily empty space where personality variation largely does not exist [31]. This interpretation would spur research into the reasons *why* these spaces are empty (e.g., it may be that few behaviors relate simultaneously to both domains).

## Domain cores and peripheries

Not all personality facets relate equally to their parent domain. Some facets are situated conceptually and empirically at a domain's core, as indicated by strong correlations with many other facets within that domain. Imagination, for example, is a core facet of openness that is correlated with most other openness facets (such as absorption and intellect), even though those facets are not correlated with each other [32]. Central facets therefore help explain patterns of covariance within a domain (e.g. people who are easily absorbed into their surroundings and those who are intellectual both tend to have vivid imaginations). Although there is general agreement on the gross features of each domain's core, different trait measures are often anchored around different core facets (for a brief review, see [10]). The abundance of personality measures in the ESCS provides a unique opportunity to identify a transtheoretical core to each Big Five domain by triangulating across personality facets from many inventories.

In contrast, some facets may be peripherally associated with a domain, as evidenced by weak correlations with the domain's other facets despite conceptual connections to the domain as a whole. For example, traditionalism may be a peripheral facet of conscientiousness that is only moderately correlated with other conscientiousness facets [33]. A clear-cut identification of peripheral facets is difficult because metrics for the boundaries between peripheral facets and facets beyond a domain remain unclear. Moreover, because different instruments measure different facets [10,34], researchers interested in comprehensively identifying domain peripheries must simultaneously examine facets from many different instruments.

Nevertheless, charting the periphery of trait domains can provide important insights into personality structure and potentially improve prediction in applied settings. Some research suggests that facet level scales are better predictors of certain outcomes than the Big Five domains ([22, 33], but see [35]). This would be particularly likely when facet scales contain specific outcome-related variance that is averaged-out when computing broader domain scores. As peripheral facets are less strongly associated with the core of a domain, they may be especially likely to contain unique variance not shared with the domain scale. Identifying the peripheral facets of a domain thus allows researchers to measure combinations of facets that contain unique variance, potentially improving the predictive power of trait models.

## Targeted understanding of particular facets or measures

Just as geneticists can use atlases to understand how different genes relate to one another, personality researchers could use a trait atlas to better understand a particular measure by examining its associations with other measures and its positioning within a domain. For example, a researcher interested in impulsivity could consult a facet atlas to investigate 1) the correlations between various impulsivity scales, 2) the extent to which impulsivity scales reflect blended content of multiple Big Five domains, and 3) whether these impulsivity scales fall in the core or periphery of each Big Five domain [36]. Together, this information may clarify cases of jingle and jangle, synthesize past research on impulsivity, and help researchers select impulsivity scales that are tuned for testing their specific hypotheses.

A facet atlas also allows researchers to test hypotheses concerning the structure of particular measures. The content of a personality measure reflects the joint effect of its creator's beliefs about personality structure and evidence regarding those beliefs [17]. For example, the Big Five Aspects Scales (BFAS) were designed to measure two maximally distinct aspects within each of the Big Five domains [12]. In essence, the BFAS presents the hypothesis that variation within each Big Five domain can be summarized by two empirically-identified facets. It follows that 1) each scale should parsimoniously cover much of the domain's space, 2) the two scales within a domain should be maximally distinctive from one another, and 3) each scale should be associated only with the domain it is intended to measure. The ESCS affords a relatively theory-neutral environment to test these structural theories because it contains scales from many personality measures.

## Examining facets using a network approach

In this study, we created a facet atlas by estimating and visualizing a series of network graphs. As the application of network modeling to the study of personality is relatively new, we offer a brief overview of this approach (for a more thorough review, see [37]). Network graphs visualize the connective patterns (e.g., correlations) among a set of variables. In personality networks, variables represented by circles (called nodes) are connected by lines (edges) that vary in width depending on the strength of correlation between the two variables. The nodes and edges are plotted in a two-dimensional space, such that nodes with similar patterns of correlation are plotted near one another, and nodes with dissimilar patterns of correlation are plotted farther apart [38]. Furthermore, nodes that correlate strongly with many other nodes are placed at the center of a network, whereas nodes with weaker correlations are placed nearer the periphery. Finally, centrality indices are computed that summarize each node's position in the network.

Although cross-sectional personality network graphs and correlation matrices are based on the same information, network graphs present complex correlation patterns spatially, highlighting relevant information and summarizing each variable's correlations with all other variables in the matrix. For instance, the core of a personality domain, which can be represented as the facets that are most strongly correlated overall with the other facets in the network, is visualized by placing strongly connected facets at the center of a network. The core or peripheral placement of a facet is thus much more easily identifiable in a network graph than in a large correlation matrix (which, in this study, would require 30 sheets of paper to display).

Network graphs also provide a complementary approach to factor analyses. Factor analysis allows researchers to distill a large covariance matrix into a smaller factor structure that captures the main dimensions of variability among items, such as the Big Five. Conversely, network graphs retain focus on the lower-level variables (in this case facets). As such, network visualizations depict individual variable-to-variable correlations, which are not the focus of factor analytic results. In sum, the workhorse of personality is the correlation matrix, and network models are a useful tool that facilitate the understanding of large, complex correlation matrices and complement factor analyses.

## The present study

In the present study, we used a network approach to examine patterns of correlation between 268 facets measured in 13 personality questionnaires that were administered to the ESCS. We estimated a network for each Big Five domain according to the results of an exploratory factor analysis of 268 facet scales. We visualized these networks to create a facet atlas, presented in this manuscript and as an online shiny app (https://tedschwaba.shinyapps.io/appdata/). We then explored how this atlas can be used to investigate blended personality facets as well as the core and periphery of each Big Five domain. Finally, we showcase how this facet atlas can be

used to better understand particular constructs by examining facets measuring impulsivity, and we illustrate how it can test structural hypotheses contained within measures by examining the Big Five Aspect Scales.

## Methods

### Participants

ESCS participants were recruited through a mailed invitation in 1993 and completed a variety of personality questionnaires sent in separate mailers beginning in 1993 (N = 1,134), with 88% retention over the following 10 years. In 1993, participants ranged in age from 18–85 (M = 49.67, SD = 13.08), and 34.7% of participants were ages 40–49. The sample was composed of 57% women, 98.4% European-Americans, and 59% of participants had at least a college degree. The ethnic and geographical homogeneity of the ESCS limit the generalizability of results, as we note in the discussion section. The number of participants varies by scale, as questionnaires were over a period of 2 decades. The items in some scales were administered over a period of multiple years, leading some facet scales within the same questionnaire to have different sample sizes.

### Measures

In this study, we included omnibus scales with either explicit hierarchical structure, hierarchical structure validated in other studies, or that measure many traits, such that each trait would approximate the scope of a facet. We present these measures in Table 1. Information for each facet scale, including N, alpha, and Big Five factor loadings, is available online at https://osf.io/f47xu/, and a correlation matrix of all facets is available at https://osf.io/w682t/.

### Analytic strategy

Analyses were conducted in R [48] using the packages psych [49] and qgraph [50]. The raw ESCS data are publicly available at https://dataverse.harvard.edu/dataverse/ESCS-Data, and the cleaned data and R scripts used in this study are available at https://osf.io/cjz8e.

**Table 1. Measures in the present study.**

| Scale | Reference | Year(s) | N | N items | N facets |
|---|---|---|---|---|---|
| Abridged Big Five Circumplex (AB5C) | [9,13] | 1994–1996 | 795–917 | 485 | 45 |
| Big Five Aspects Scales (BFAS) | [12] | 1994–1996 | 905–945 | 98[a] | 10 |
| Big Five Inventory (BFI) | [39] | 1998 | 703 | 35 | 10 |
| Behavioral Inhibition System/Behavioral Approach System Scales (BIS/BAS) | [40] | 2003 | 734 | 20 | 4 |
| California Personality Inventory (CPI) | [14] | 1994 | 792 | 462 | 20 |
| HEXACO Personality Inventory | [8] | 2003 | 737 | 192 | 24 |
| Hogan Personality Inventory (HPI) | [41] | 1997 | 739–742 | 193 | 35[b] |
| Jackson Personality Inventory-Revised (JPI-R) | [42] | 1999 | 712 | 300 | 15 |
| Multidimensional Personality Questionnaire (MPQ) | [43] | 1999 | 733 | 276 | 11 |
| NEO Personality Inventory-Revised (NEO-PI-R) | [44] | 1994 | 857 | 240 | 30 |
| Six Factor Personality Questionnaire (6FPQ) | [45] | 1999 | 714 | 108 | 18 |
| Sixteen Personality Factor Questionnaire, Fifth Edition (16PF) | [46] | 1996 | 680 | 185 | 16 |
| Temperament and Character Inventory | [47] | 1997 | 727 | 295 | 30 |

[a] = The BFAS scale includes 100 items total, 2 of which were not measured in the ESCS.

[b] = In this study, we omitted nine HPI facets (science ability, intellectual games, education, math ability, good memory, reading, self-focus, impression management, appearance) that measured ability, as these scales did not correlate highly with other facet scales.

Before visualizing this facet atlas, we first organized all 268 ESCS facets into the Big Five domains using exploratory factor analysis with oblimin rotation. All facets with factor loadings greater than |.25| were included in that domain. This threshold was chosen after reviewing the results of the factor analysis because it offered the best balance between our desires to include some facets in multiple domains and exclude facets with minimal conceptual and empirical association to a given domain. Two facets, HPI Not Autonomous and HPI Not Spontaneous, did not load .25 on any domain and thus were not included in any network. This .25 factor loading threshold can be changed in the online app.

For each Big Five domain, we then estimated a full (rather than partial) correlation matrix, using pairwise deletion, which described associations among facets. Although psychological network research often uses partial correlation methods [51], we estimated full correlation matrices for each network. A partial correlation estimates the association between a predictor and outcome while controlling for all other predictor-outcome associations in the network. In the present case, many scales in each network measured the same, or highly similar constructs (e.g. there were multiple sociability scales in the extraversion network) As such, the partial correlation between CPI *sociability* and HPI *likes parties* would be estimated while controlling for HEXACO *sociability*, removing important sociability-related variance and rendering results uninterpretable (see [52]). Controlling for this variance also removes meaningful patterns of correlation that arise from common latent factors (i.e. the Big Five, [53]) and makes the overall network structure less stable and replicable [54]. Thus, for this study, bivariate correlations were more appropriate than partial correlations.

We also corrected all associations for measurement error. Measurement error deflates correlations measured with lower reliability and is partly a function of instrument length, such that shorter instruments are typically less reliable [55]. Because the facet scales in this study varied in length from 1 to 46 items, it was especially important to correct for measurement error due to scale length. As we lacked item-level information for some scales, we applied a rough correction for unreliability using Cronbach's alpha [56]; see http://ipip.ori.org/newMultipleconstructs.htm for a list of scale alphas calculated using item-level ESCS data). This correction for unreliability can be toggled in the online app. Because some ESCS scales were administered years apart from one another, we note that some correlations may remain somewhat attenuated (see [57]).

Finally, we estimated the most central and peripheral facets for each Big Five domain by calculating each facet's network strength centrality. Strength is the sum of the absolute values of all correlations that each facet has with all other facets in a network [58]. In this study we calculate strength as the average of all correlations so that this metric is comparable across Big Five domains and with other networks. Facets with higher strength are relatively strongly associated with the other facets in a network, so we consider them to be core. This approach to defining the core of a network is conceptually similar to previous research that has argued that core facets have the highest loadings on a latent domain factor (e.g., [59]), and we examine if this is empirically true, as well. The converse logic applies to peripheral facets. Despite substantial factor loadings on the overall Big Five domain, peripheral facets have low network strength and are weakly associated with the other facets in a domain.

Past research has highlighted the importance of estimating centrality index reliability [37]; although this is less of an issue when estimating networks from full correlations [54]. To construct 95% confidence intervals for each facet's strength, we adapted code from the bootnet package [37] and simulated 1,000 bootstrapped iterations of each network. This code is available at https://osf.io/9j3pm/.

## Results

### Network visualization and shiny app

In Figs 1–5, we present a facet atlas, which displays the intercorrelations between the facets in each Big Five domain in rich network-based visualizations. For more information, we offer a point-and-click online app, hosted at https://tedschwaba.shinyapps.io/appdata/. In this app, users can explore a full network of all 268 facets, adjust the threshold for blended facets, and toggle the correction for measurement error. The app also displays complete descriptive information for all facet scales and allows users to visualize networks in terms of strength centrality. In the following sections, we describe how this facet atlas can be used to address major questions about the lower-level structure of personality outlined in the introduction.

### Blended facets

Results of the factor analysis suggested that the majority of facets (157 of 268, or 58.5%) contained a blended loading pattern with multiple factor loadings greater than |.25| (excluding the 40 AB5C facets designed to blended, 55.3% of facets were blended). Furthermore, 18 facets

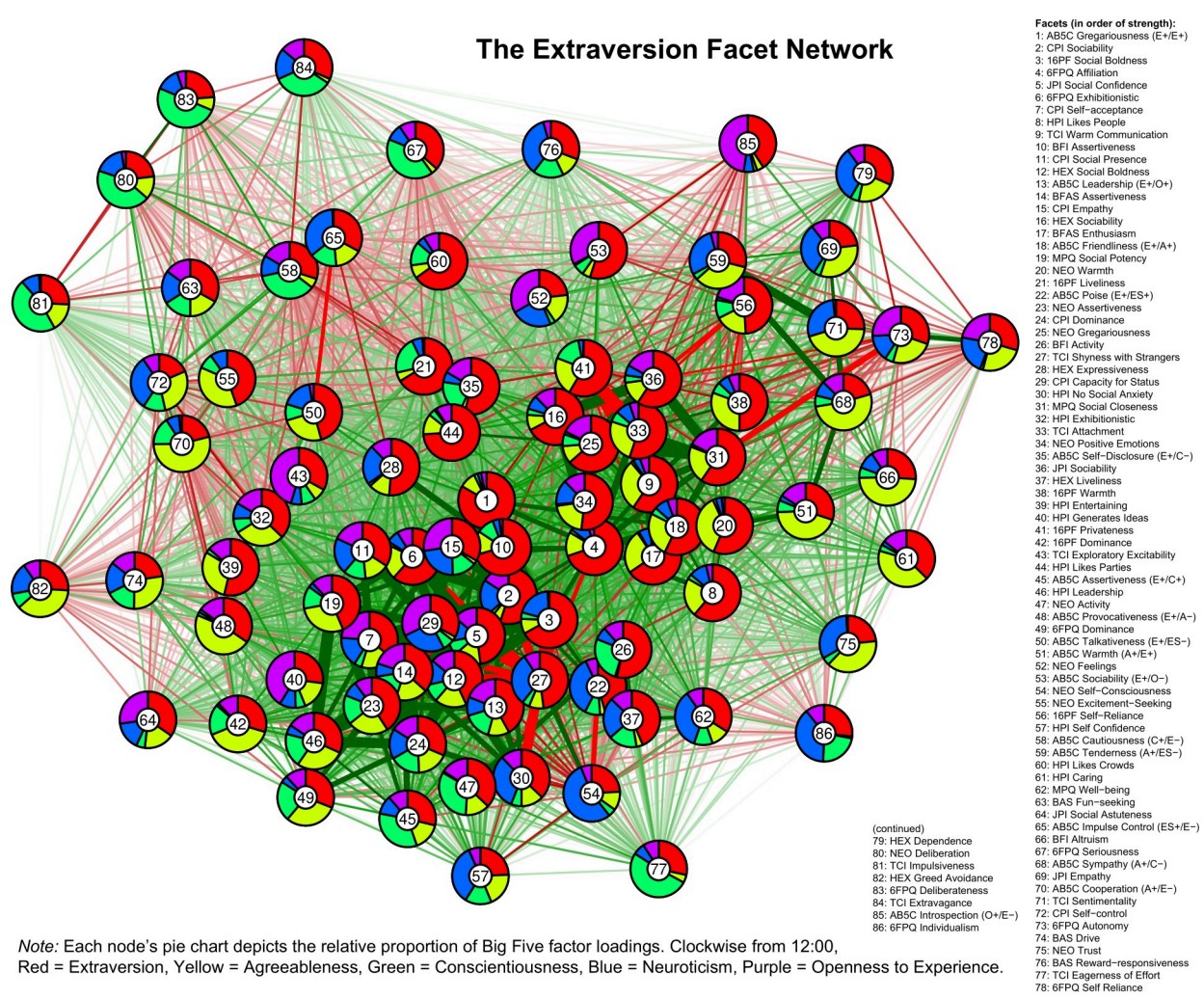

**The Extraversion Facet Network**

**Facets (in order of strength):**
1: AB5C Gregariousness (E+/E+)
2: CPI Sociability
3: 16PF Social Boldness
4: 6FPQ Affiliation
5: JPI Social Confidence
6: 6FPQ Exhibitionistic
7: CPI Self−acceptance
8: HPI Likes People
9: TCI Warm Communication
10: BFI Assertiveness
11: CPI Social Presence
12: HEX Social Boldness
13: AB5C Leadership (E+/O+)
14: BFAS Assertiveness
15: CPI Empathy
16: HEX Sociability
17: BFAS Enthusiasm
18: AB5C Friendliness (E+/A+)
19: MPQ Social Potency
20: NEO Warmth
21: 16PF Liveliness
22: AB5C Poise (E+/ES+)
23: NEO Assertiveness
24: CPI Dominance
25: NEO Gregariousness
26: BFI Activity
27: TCI Shyness with Strangers
28: HEX Expressiveness
29: CPI Capacity for Status
30: HPI No Social Anxiety
31: MPQ Social Closeness
32: HPI Exhibitionistic
33: TCI Attachment
34: NEO Positive Emotions
35: AB5C Self−Disclosure (E+/C−)
36: JPI Sociability
37: HEX Liveliness
38: 16PF Warmth
39: HPI Entertaining
40: HPI Generates Ideas
41: 16PF Privateness
42: 16PF Dominance
43: TCI Exploratory Excitability
44: HPI Likes Parties
45: AB5C Assertiveness (E+/C+)
46: HPI Leadership
47: NEO Activity
48: AB5C Provocativeness (E+/A−)
49: 6FPQ Dominance
50: AB5C Talkativeness (E+/ES−)
51: AB5C Warmth (A+/E+)
52: NEO Feelings
53: AB5C Sociability (E+/O−)
54: NEO Self−Consciousness
55: NEO Excitement−Seeking
56: 16PF Self−Reliance
57: HPI Self Confidence
58: AB5C Cautiousness (C+/E−)
59: AB5C Tenderness (A+/ES−)
60: HPI Likes Crowds
61: HPI Caring
62: MPQ Well−being
63: BAS Fun−seeking
64: JPI Social Astuteness
65: AB5C Impulse Control (ES+/E−)
66: BFI Altruism
67: 6FPQ Seriousness
68: AB5C Sympathy (A+/C−)
69: JPI Empathy
70: AB5C Cooperation (A+/E−)
71: TCI Sentimentality
72: CPI Self−control
73: 6FPQ Autonomy
74: BAS Drive
75: NEO Trust
76: BAS Reward−responsiveness
77: TCI Eagerness of Effort
78: 6FPQ Self Reliance

(continued)
79: HEX Dependence
80: NEO Deliberation
81: TCI Impulsiveness
82: HEX Greed Avoidance
83: 6FPQ Deliberateness
84: TCI Extravagance
85: AB5C Introspection (O+/E−)
86: 6FPQ Individualism

*Note:* Each node's pie chart depicts the relative proportion of Big Five factor loadings. Clockwise from 12:00, Red = Extraversion, Yellow = Agreeableness, Green = Conscientiousness, Blue = Neuroticism, Purple = Openness to Experience.

**Fig 1. Extraversion facet network.**

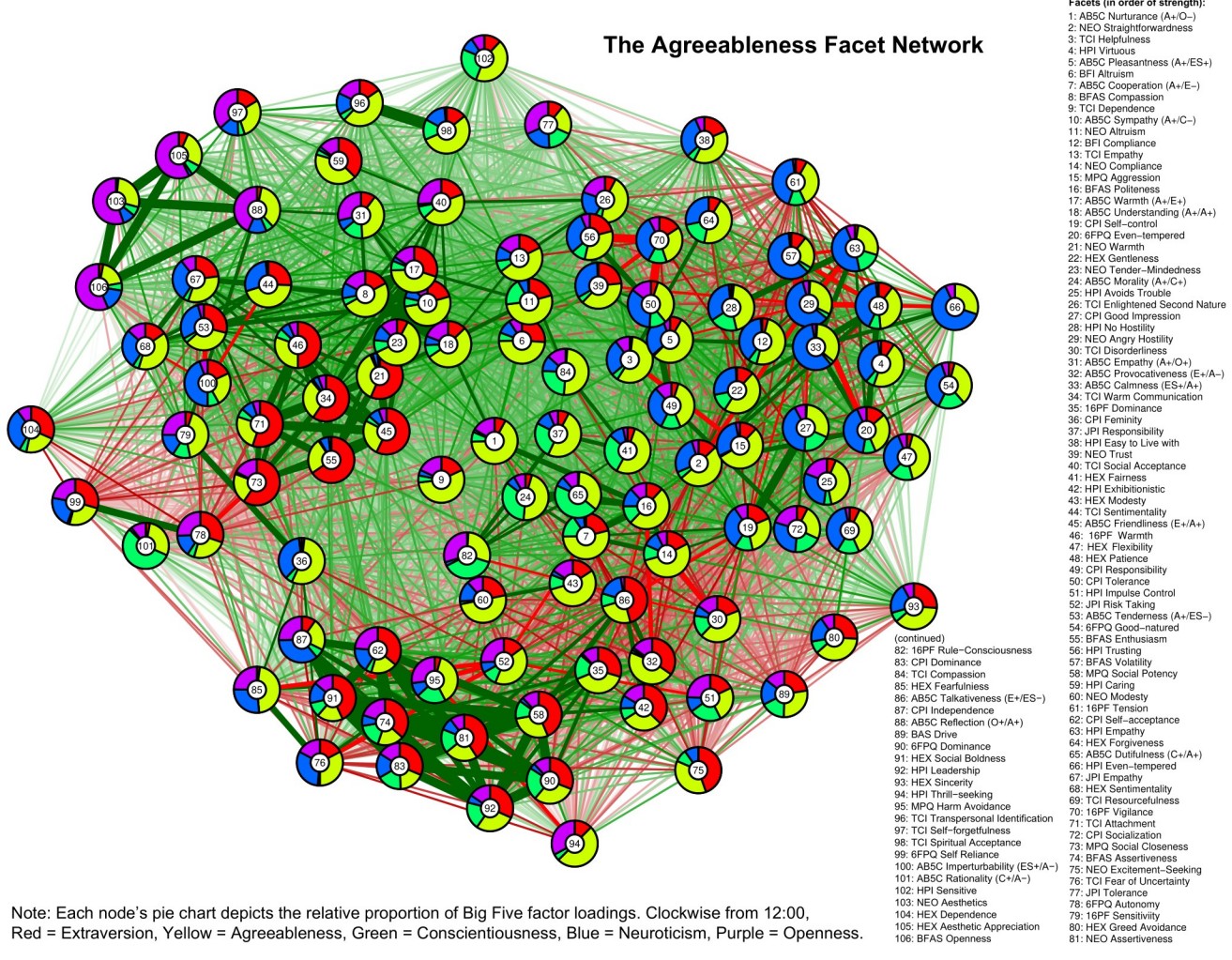

**The Agreeableness Facet Network**

**Facets (in order of strength):**
1: AB5C Nurturance (A+/O–)
2: NEO Straightforwardness
3: TCI Helpfulness
4: HPI Virtuous
5: AB5C Pleasantness (A+/ES+)
6: BFI Altruism
7: AB5C Cooperation (A+/E–)
8: BFAS Compassion
9: TCI Dependence
10: AB5C Sympathy (A+/C–)
11: NEO Altruism
12: BFI Compliance
13: TCI Empathy
14: NEO Compliance
15: MPQ Aggression
16: BFAS Politeness
17: AB5C Warmth (A+/E+)
18: AB5C Understanding (A+/A+)
19: CPI Self–control
20: 6FPQ Even–tempered
21: NEO Warmth
22: HEX Gentleness
23: NEO Tender–Mindedness
24: AB5C Morality (A+/C+)
25: HPI Avoids Trouble
26: TCI Enlightened Second Nature
27: CPI Good Impression
28: HPI No Hostility
29: NEO Angry Hostility
30: TCI Disorderliness
31: AB5C Empathy (A+/O+)
32: AB5C Provocativeness (E+/A–)
33: AB5C Calmness (ES+/A+)
34: TCI Warm Communication
35: 16PF Dominance
36: CPI Feminity
37: JPI Responsibility
38: HPI Easy to Live with
39: NEO Trust
40: TCI Social Acceptance
41: HEX Fairness
42: HPI Exhibitionistic
43: HEX Modesty
44: TCI Sentimentality
45: AB5C Friendliness (E+/A+)
46: 16PF Warmth
47: HEX Flexibility
48: HEX Patience
49: CPI Responsibility
50: CPI Tolerance
51: HPI Impulse Control
52: JPI Risk Taking
53: AB5C Tenderness (A+/ES+)
54: 6FPQ Good–natured
55: BFAS Enthusiasm
56: HPI Trusting
57: BFAS Volatility
58: MPQ Social Potency
59: HPI Caring
60: NEO Modesty
61: 16PF Tension
62: CPI Self–acceptance
63: HPI Empathy
64: HEX Forgiveness
65: AB5C Dutifulness (C+/A+)
66: HPI Even–tempered
67: JPI Empathy
68: HEX Sentimentality
69: TCI Resourcefulness
70: 16PF Vigilance
71: TCI Attachment
72: CPI Socialization
73: MPQ Social Closeness
74: BFAS Assertiveness
75: NEO Excitement–Seeking
76: TCI Fear of Uncertainty
77: JPI Tolerance
78: 6FPQ Autonomy
79: 16PF Sensitivity
80: HEX Greed Avoidance
81: NEO Assertiveness

(continued)
82: 16PF Rule–Consciousness
83: CPI Dominance
84: TCI Compassion
85: HEX Fearfulness
86: AB5C Talkativeness (E+/ES–)
87: CPI Independence
88: AB5C Reflection (O+/A+)
89: BAS Drive
90: 6FPQ Dominance
91: HEX Social Boldness
92: HPI Leadership
93: HEX Sincerity
94: HPI Thrill–seeking
95: MPQ Harm Avoidance
96: TCI Transpersonal Identification
97: TCI Self–forgetfulness
98: TCI Spiritual Acceptance
99: 6FPQ Self Reliance
100: AB5C Imperturbability (ES+/A–)
101: AB5C Rationality (C+/A–)
102: HPI Sensitive
103: NEO Aesthetics
104: HEX Dependence
105: HEX Aesthetic Appreciation
106: BFAS Openness

Note: Each node's pie chart depicts the relative proportion of Big Five factor loadings. Clockwise from 12:00,
Red = Extraversion, Yellow = Agreeableness, Green = Conscientiousness, Blue = Neuroticism, Purple = Openness.

**Fig 2. Agreeableness facet network.**

displayed blends with three domains, two facets displayed blends with four domains (BFAS *assertiveness* and J6F *self-reliance*) and one facet displayed blends with all five domains (CPI *dominance*). Blends with agreeableness (88 blends) were most common, followed by neuroticism (81), extraversion (76), openness (52) and conscientiousness (48). We summarize these patterns in Table 2 and visualize each facet's blendedness in Figs 1–5.

Facets with similar patterns of blendedness tended to cluster together, meaning that they had similar patterns of correlation with other facets in the network. For example, the top of the extraversion network contained a group of facets that were blends of extraversion, agreeableness, and low neuroticism. Positive and contrasting blends generally repulsed each other within a network, illustrating circumplex-like constellations. For example, the bottom-left side of the openness network contained facets with a contrasting blend of high openness and low agreeableness (such as 6FPQ *autonomy*), but the upper-right of the openness network contained facets with a positive blend of high openness and high agreeableness (such as 16PF *sensitivity*).

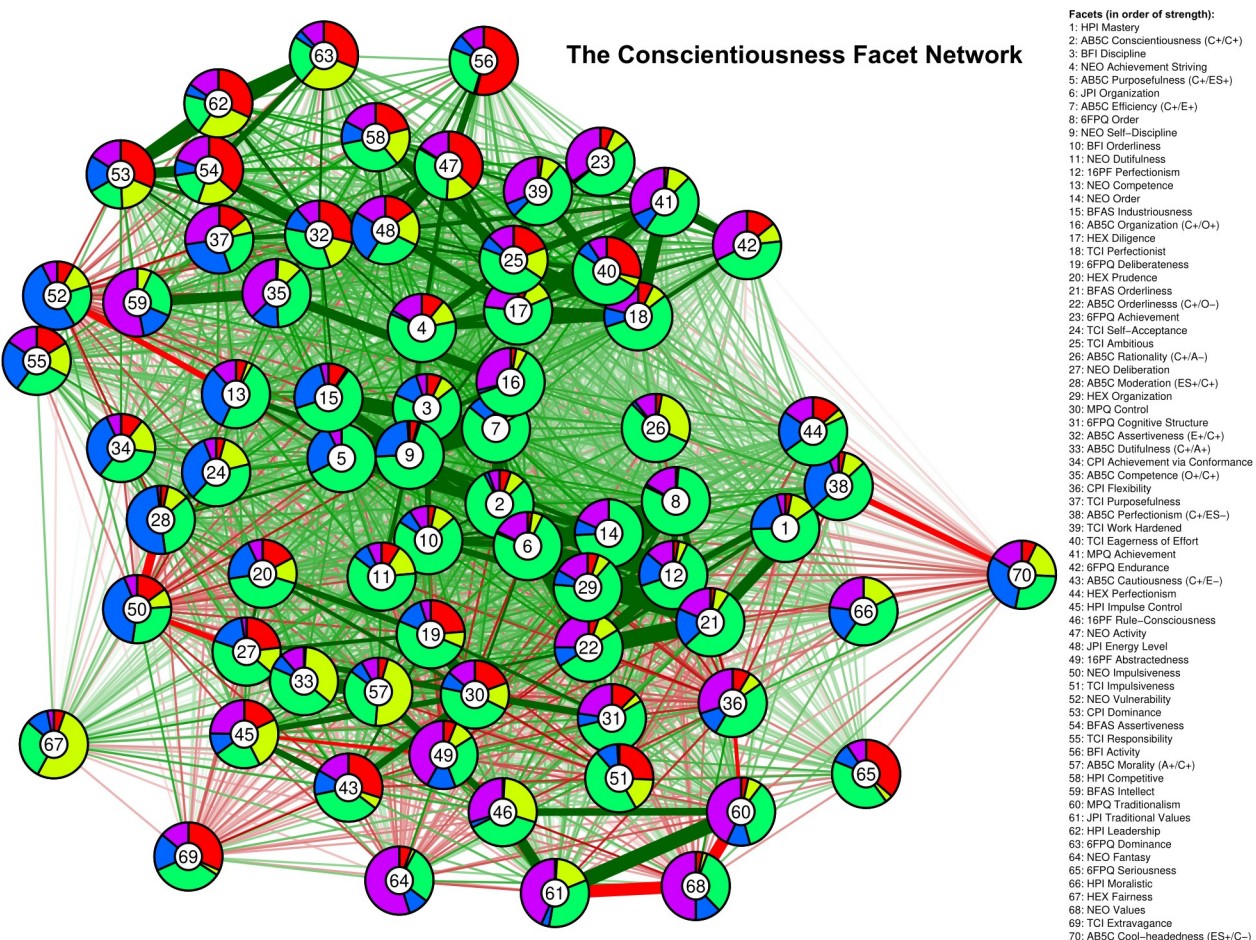

**The Consientiousness Facet Network**

**Facets (in order of strength):**
1: HPI Mastery
2: AB5C Conscientiousness (C+/C+)
3: BFI Discipline
4: NEO Achievement Striving
5: AB5C Purposefulness (C+/ES+)
6: JPI Organization
7: AB5C Efficiency (C+/E+)
8: 6FPQ Order
9: NEO Self–Discipline
10: BFI Orderliness
11: NEO Dutifulness
12: 16PF Perfectionism
13: NEO Competence
14: NEO Order
15: BFAS Industriousness
16: AB5C Organization (C+/O+)
17: HEX Diligence
18: TCI Perfectionist
19: 6FPQ Deliberateness
20: HEX Prudence
21: BFAS Orderliness
22: AB5C Orderlinesss (C+/O–)
23: 6FPQ Achievement
24: TCI Self–Acceptance
25: TCI Ambitious
26: AB5C Rationality (C+/A–)
27: NEO Deliberation
28: AB5C Moderation (ES+/C+)
29: HEX Organization
30: MPQ Control
31: 6FPQ Cognitive Structure
32: AB5C Assertiveness (E+/C+)
33: AB5C Dutifulness (C+/A+)
34: CPI Achievement via Conformance
35: AB5C Competence (O+/C+)
36: CPI Flexibility
37: TCI Purposefulness
38: AB5C Perfectionism (C+/ES–)
39: TCI Work Hardened
40: TCI Eagerness of Effort
41: MPQ Achievement
42: 6FPQ Endurance
43: AB5C Cautiousness (C+/E–)
44: HEX Perfectionism
45: HPI Impulse Control
46: 16PF Rule–Consciousness
47: NEO Activity
48: JPI Energy Level
49: 16PF Abstractedness
50: NEO Impulsiveness
51: TCI Impulsiveness
52: NEO Vulnerability
53: CPI Dominance
54: BFAS Assertiveness
55: TCI Responsibility
56: BFI Activity
57: AB5C Morality (A+/C+)
58: HPI Competitive
59: BFAS Intellect
60: MPQ Traditionalism
61: JPI Traditional Values
62: HPI Leadership
63: 6FPQ Dominance
64: NEO Fantasy
65: 6FPQ Seriousness
66: HPI Moralistic
67: HEX Fairness
68: NEO Values
69: TCI Extravagance
70: AB5C Cool–headedness (ES+/C–)

Note: Each node's pie chart depicts the relative proportion of Big Five factor loadings. Clockwise from 12:00,
Red = Extraversion, Yellow = Agreeableness, Green = Conscientiousness, Blue = Neuroticism, Purple = Openness.

**Fig 3. Conscientiousness facet network.**

### Core and peripheral facets

In Figs 1–5, nodes are labeled according to their strength, ranked from strongest to weakest. In Figs 6–10, we visualize network strength for each network's 10 most core and 10 most peripheral facets. Strength estimates for all facets in each network are available at https://osf.io/cjz8e. Strength centrality estimates were highly correlated with the absolute value of factor loadings on that domain. Correlations were .82 for extraversion, .76 for agreeableness, .86 for conscientiousness, .79 for neuroticism, and .71 for openness

As can be seen in these figures, there were large differences in strength between the most peripheral facets and the most core facets. However, within each network's core, no single scale or construct stood out as being most central. For example, within the set of the strongest agreeableness facets were scales measuring interpersonal warmth, altruism, and compassion, three related but distinct constructs. Possible exceptions were the neuroticism network, with core facets representing emotional reactivity and anxiety [12] and the extraversion network, with core facets primarily representing sociability [59].

The periphery of each domain's network was also heterogenous in content and often contained blended facets. For example, the periphery of the conscientiousness network was

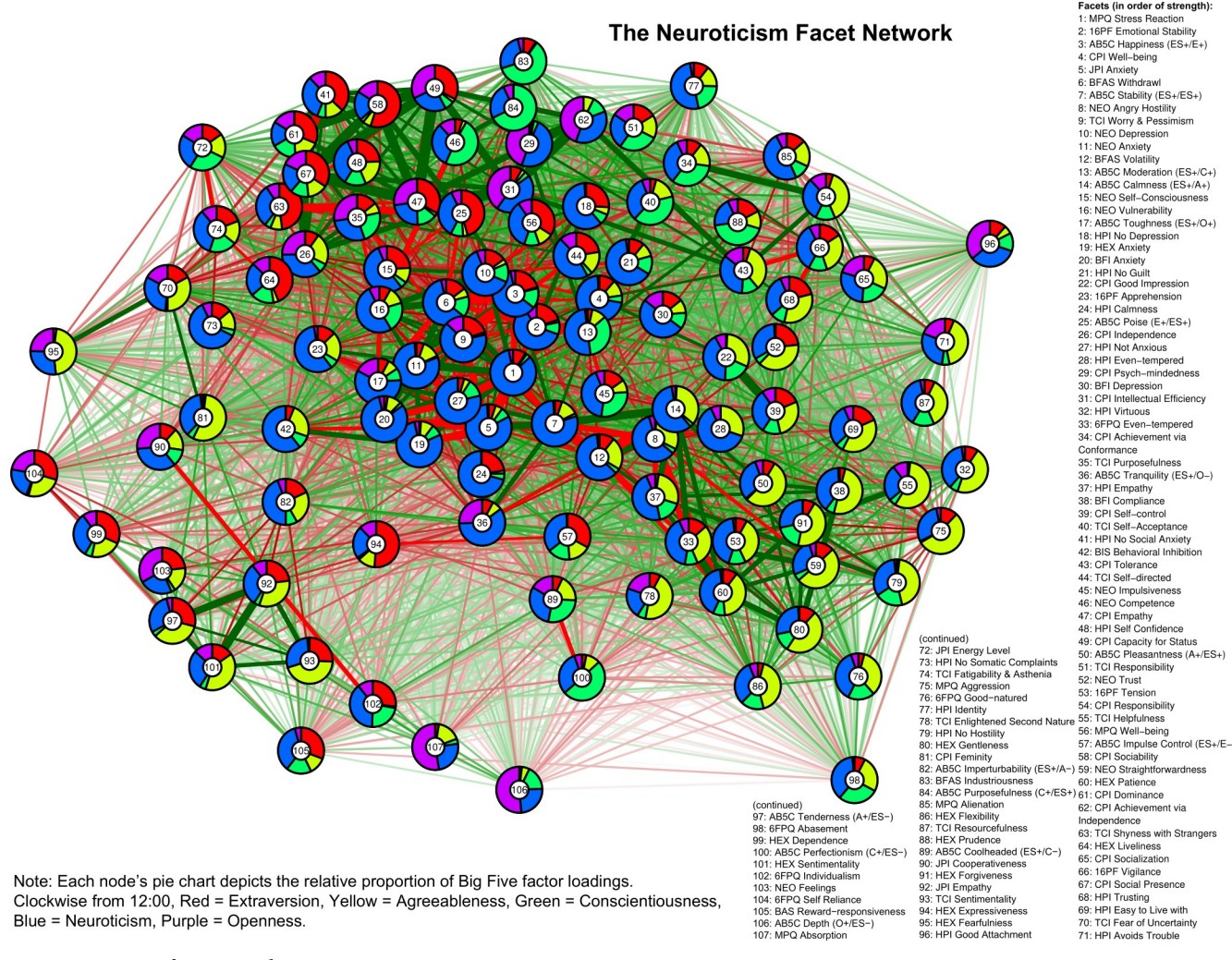

**The Neuroticism Facet Network**

Note: Each node's pie chart depicts the relative proportion of Big Five factor loadings.
Clockwise from 12:00, Red = Extraversion, Yellow = Agreeableness, Green = Conscientiousness,
Blue = Neuroticism, Purple = Openness.

**Fig 4. Neuroticism facet network.**

composed of facets measuring diverse constructs including values (blended with openness), activity (blended with extraversion), and leadership (blended with extraversion and agreeableness). Overall, few facets appeared to be in conceptually unrelated networks where they did not belong, supporting our decision to use a factor loading of >|.25| for domain inclusion.

## Impulsivity in this facet atlas

This facet atlas can be used as a practical resource to understand a single measure or construct in more depth. We illustrate this by examining impulsivity. In the ESCS, four facet scales are labeled as some variant of impulsivity: NEO *impulsiveness*, TCI *impulsiveness*, AB5C *impulse control*, and HPI *impulse control*. These four scales had intercorrelations that ranged from .22 to .44 (Table 3), indicating that these scales measure a heterogenous set of constructs [36]. Accordingly, each impulsivity scale had a different blend of factor loadings: AB5C *impulse control* appeared in the neuroticism (factor loading = -.49) and extraversion (-.42) networks, NEO *impulsiveness* appeared in the neuroticism (.47) and conscientiousness (-.32) networks, HPI *impulse control* appeared in the agreeableness (.33), conscientiousness (.30) and openness (-.33) networks, and TCI *impulsiveness* appeared in the extraversion (.27) and

**The Openness Facet Network**

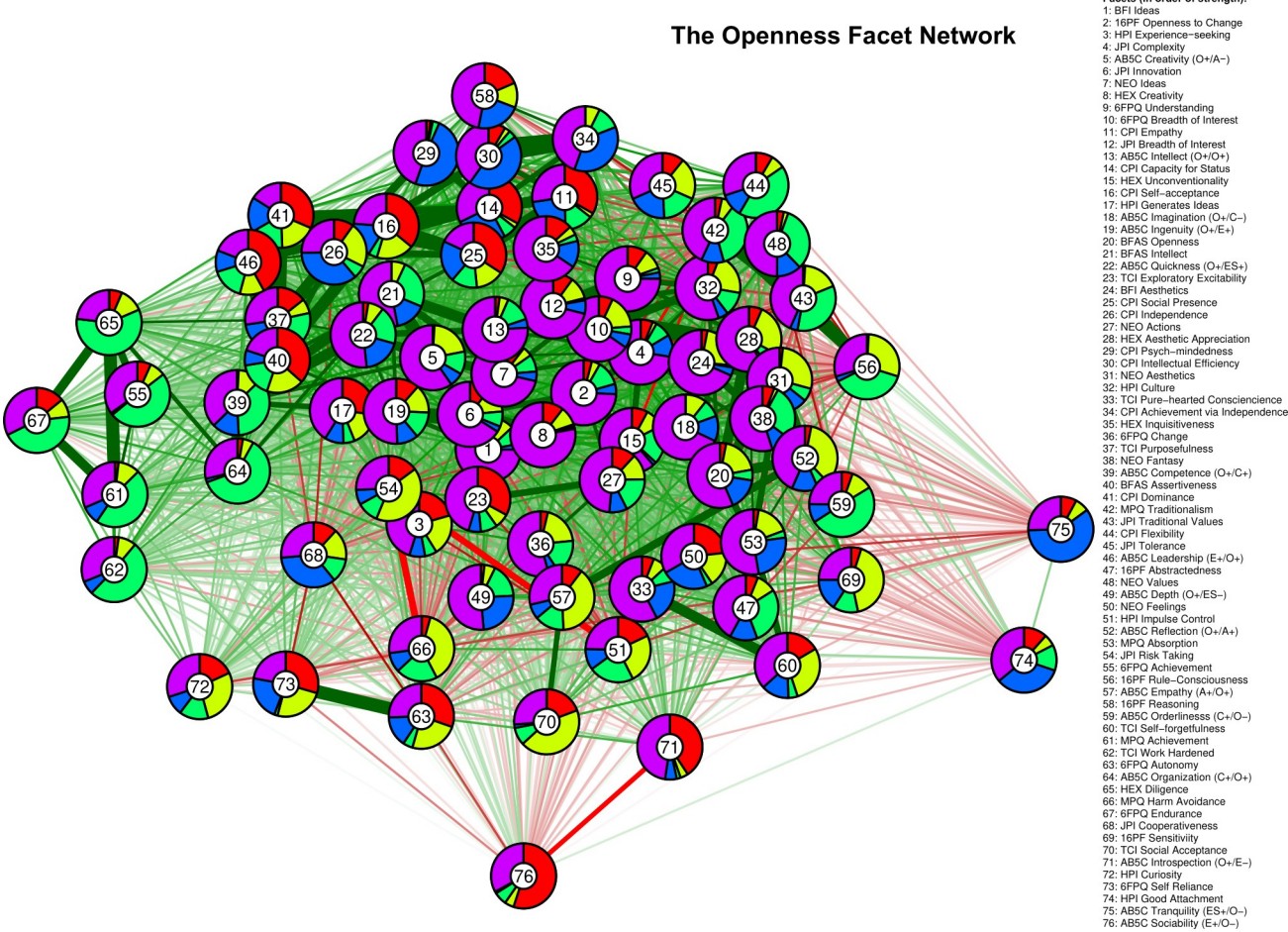

Facets (in order of strength):
1: BFI Ideas
2: 16PF Openness to Change
3: HPI Experience-seeking
4: JPI Complexity
5: AB5C Creativity (O+/A−)
6: JPI Innovation
7: NEO Ideas
8: HEX Creativity
9: 6FPQ Understanding
10: 6FPQ Breadth of Interest
11: CPI Empathy
12: JPI Breadth of Interest
13: AB5C Intellect (O+/O+)
14: CPI Capacity for Status
15: HEX Unconventionality
16: CPI Self-acceptance
17: HPI Generates Ideas
18: AB5C Imagination (O+/C−)
19: AB5C Ingenuity (O+/E+)
20: BFAS Openness
21: BFAS Intellect
22: AB5C Quickness (O+/ES+)
23: TCI Exploratory Excitability
24: BFI Aesthetics
25: CPI Social Presence
26: NEO Actions
27: NEO Aesthetics
28: HEX Aesthetic Appreciation
29: CPI Psych-mindedness
30: CPI Intellectual Efficiency
31: NEO Aesthetics
32: HPI Culture
33: TCI Pure-hearted Consciencience
34: CPI Achievement via Independence
35: HEX Inquisitiveness
36: 6FPQ Change
37: TCI Purposefulness
38: NEO Fantasy
39: AB5C Competence (O+/C+)
40: BFAS Assertiveness
41: CPI Dominance
42: MPQ Traditionalism
43: JPI Traditional Values
44: CPI Flexibility
45: JPI Tolerance
46: AB5C Leadership (E+/O+)
47: 16PF Abstractedness
48: NEO Values
49: AB5C Depth (O+/ES−)
50: NEO Feelings
51: HPI Impulse Control
52: AB5C Reflection (O+/A+)
53: MPQ Absorption
54: JPI Risk Taking
55: 6FPQ Achievement
56: 16PF Rule-Consciousness
57: AB5C Empathy (A+/O+)
58: 16PF Reasoning
59: AB5C Orderlinesss (C+/O−)
60: TCI Self-forgetfulness
61: MPQ Achievement
62: TCI Work Hardened
63: 6FPQ Autonomy
64: AB5C Organization (C+/O+)
65: HEX Diligence
66: MPQ Harm Avoidance
67: 6FPQ Endurance
68: JPI Cooperativeness
69: 16PF Sensitivity
70: TCI Social Acceptance
71: AB5C Introspection (O+/E−)
72: HPI Curiosity
73: 6FPQ Self Reliance
74: HPI Good Attachment
75: AB5C Tranquility (ES+/O−)
76: AB5C Sociability (E+/O−)

Note: Each node's pie chart depicts the relative proportion of Big Five factor loadings.
Clockwise from 12:00, Red = Extraversion, Yellow = Agreeableness, Green = Conscientiousness, Blue = Neuroticism, Purple = Openness.

**Fig 5. Openness facet network.**

conscientiousness (-.49) networks. This suggests that each of the scales besides AB5C *impulse control* measures conscientiousness-related features, and AB5C *impulse control* and NEO *impulsiveness* measure clinically relevant neuroticism-related features.

We turned to this facet atlas to better understand how these impulsivity scales relate to conscientiousness. In the conscientiousness network, HPI *impulse control* (strength = 22.15, 95% CI [19.77, 24.41]), NEO *impulsiveness* (strength = 21.34, 95% CI [18.49, 24.32]) and TCI *impulsiveness* (strength = 21.22, 95% CI [18.69, 23.66]) occupied similar, peripheral network positions, ranked 45th, 50th, and 51st in network strength out of 70 facets, respectively. Researchers interested in predicting conscientiousness-related outcomes may therefore benefit from measuring impulsiveness using either of these three scales, as they may contribute additional predictive variance compared to core conscientiousness scales. This facet atlas also provides insight into the scales' content. HPI *impulse control* was located near conscientiousness facets that measure deliberation, dutifulness, and cautiousness, suggesting this scale emphasizes the behavioral constraint components of impulsivity [36]. NEO *impulsiveness* was located near facets that measure moderation and prudence, suggesting that this scale emphasizes sensation-seeking components of impulsivity [36]. Finally, TCI *impulsiveness* was located near facets that measure flexibility and traditionalism, suggesting that this scale emphasizes

**Table 2. Summary of blended facet scales.**

| Blend | Number of scales | Example facets |
|---|---|---|
| E, A | 22 | NEO-PI-R Warmth, HEXACO Social Boldness |
| E, C | 10 | AB5C Cautiousness, NEO-PI-R Activity |
| E, N | 12 | TCI Shyness with Strangers, MPQ Well-Being |
| E, O | 5 | AB5C Introspection, TCI Exploratory Excitability |
| A, C | 4 | HEXACO Fairness, AB5C Rationality |
| A, N | 35 | BFAS Volatility, BFI Compliance |
| A, O | 11 | 16PFQ Sensitivity, JPI-R Risk Taking |
| C, N | 13 | NEO-PI-R Impulsiveness, BFAS Orderliness |
| C, O | 15 | MPQ Achievement, CPI Flexibility |
| N, O | 8 | CPI Psychological-Mindedness, AB5C Tranquility |
| E, A, C | 2 | J6F Dominance, HPI Leadership |
| E, A, N | 6 | NEO-PI-R Trust, HEXACO dependency |
| E, A, O | 2 | CPI Self-Acceptance, J6F Autonomy |
| E, N, O | 4 | CPI Capacity for Status, NEO Feelings |
| A, C, O | 2 | 16PFQ Rule-Consciousness, HPI Impulse Control |
| A, N, O | 1 | CPI Independence |
| C, N, O | 1 | TCI Purposefulness |

Blended facets = factor loadings > |.25| on multiple factors. E = Extraversion. A = Agreeableness.
C = Conscientiousness. N = Neuroticism. O = Openness. NEO-PI-R = NEO-Personality Inventory-Revised.
AB5C = Abridged Big Five Circumplex. TCI = Temperament and Character Inventory. MPQ = Multiphasic
Personality Questionnaire. BFAS = Big Five Aspects Scales. BFI = Big Five Inventory. 16PFQ = 16 Personality Factors
Questionnaire. JPI = Jackson Personality Inventory-Revised. J6F = Jackson Six Factor Questionnaire. HPI = Hogan
Personality Inventory. CPI = California Personality Inventory.

components of impulsivity broadly related to rule-following and norm adherence. These differences indicate a jingle issue, where three distinct constructs are being operationalized as impulsiveness. Researchers interested in studying impulsiveness may benefit from considering these components separately or in tandem [60], and from paying close attention to the scales used in past research on impulsiveness.

The two impulsiveness scales with substantial loadings on neuroticism, AB5C *impulse control* (strength = 32.23, 95% CI [28.49, 36.13]) and NEO *impulsiveness* (strength = 34.49, 95% CI [30.05, 38.61]) occupied an intermediate space between the core and periphery of neuroticism, ranked 57th and 42nd in strength out of 107 facets. This indicates that impulsiveness as measured by these scales is moderately related to other neuroticism facets. AB5C *impulse control* and NEO *impulsiveness* also occupied similar network positions in the neuroticism network, demonstrating that these scales measure similar neuroticism-related content, despite their moderate correlation ($r = .44$). Nearby neuroticism facets measure constructs such as volatility, angry hostility, and (low) cool-headedness, indicating that these two scales include components of both general affective reactivity and quickness to anger [61], which appears to be omitted from the TCI *impulsiveness* scale and HPI *impulse control* scale.

## The BFAS in this facet atlas

A facet atlas can be used to evaluate the implicit theories of trait structure inherent in most personality instruments. We illustrate this feature by examining the network placement of facets measured in the Big Five Aspects Scales (BFAS; [12]). We note that the BFAS was created by factor-analyzing the AB5C and NEO-PI-R scales in the ESCS dataset. However, the present

**Table 3. Correlations between measures of impulsivity.**

|  | 1 | 2 | 3 | 4 |
|---|---|---|---|---|
| 1) AB5C Impulse Control |  | .43 | -.65 | -.34 |
| 2) HPI Impulse Control | .32 |  | -.48 | -.48 |
| 3) NEO PI-R Impulsiveness | -.44 | -.32 |  | .33 |
| 4) TCI Impulsiveness | -.26 | -.35 | .22 |  |

N ranges from 727 to 900 across measures. All correlations are significant at $p < .001$. Correlations above the diagonal are corrected for measurement error using alpha. AB5C = Abridged Big Five circumplex. HPI = Hogan Personality Inventory. NEO PI-R = NEO Personality Inventory–Revised. TCI = Temperament and Character Inventory.

analyses include correlational patterns with 191 additional facet scales, revealing new information about how the BFAS scales measure personality traits.

Because the BFAS was created by factor analysis, the two aspect scales for each of the Big Five should both parsimoniously cover much of the domain's space. This can be evaluated by examining whether each domain's aspect scales fall within its core. This was the case for each of the five domains (see Figs 1–6). The most peripheral aspect scale was *orderliness* (ranked 21st of all 70 facet scales in the conscientiousness network). Each domain's aspect scales did not differ significantly in strength from one another, suggesting that both scales were equally core (i.e. no aspect was "more important" than the other).

A consequence of the specific factor-analytic procedure used to derive the BFAS is that, because the two aspect scales in each domain were rotated to be orthogonal, they should each measure distinctly different content. This can be evaluated by examining how far apart the two aspects are within each domain's network, as scales that have similar patterns of correlations with other facets will occupy network positions close to one another. Results indicated that the two BFAS aspect scales in each domain were indeed placed relatively far apart from one

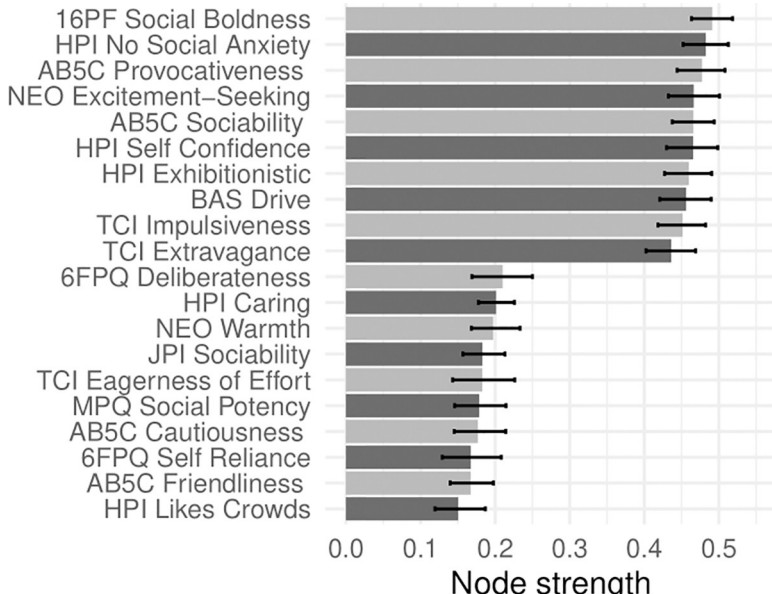

**Fig 6. The most strongly and weakly connected facets in the extraversion network.**

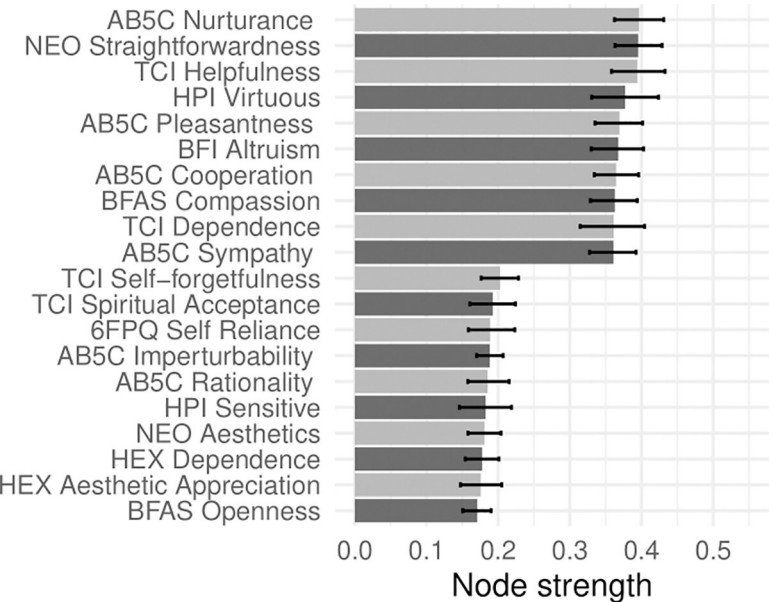

**Fig 7. The most strongly and weakly connected facets in the agreeableness network.**

another (while still remaining in the domain's core), supporting the idea that these scales cover a relatively broad content area within each domain.

We also explored the extent to which BFAS aspect scales contained blended content from multiple Big Five domains. Results indicated that six out of 10 aspect scales were exclusively associated with a single domain, whereas four scales (*assertiveness*, *industriousness*, *intellect*, and *openness*) were associated with multiple domains at a cutoff of |.25|. Most notably, BFAS *assertiveness* was a blend of four Big Five domains: extraversion, (low) agreeableness, openness,

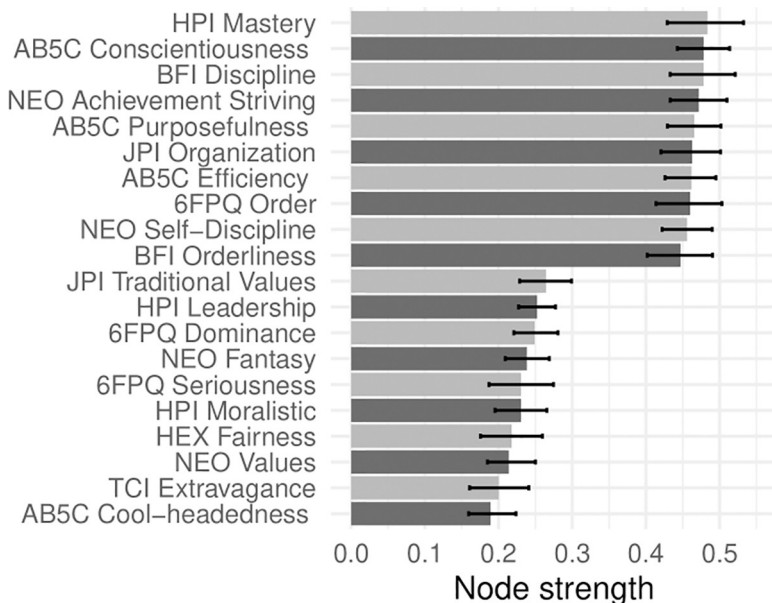

**Fig 8. The most strongly and weakly connected facets in the conscientiousness network.**

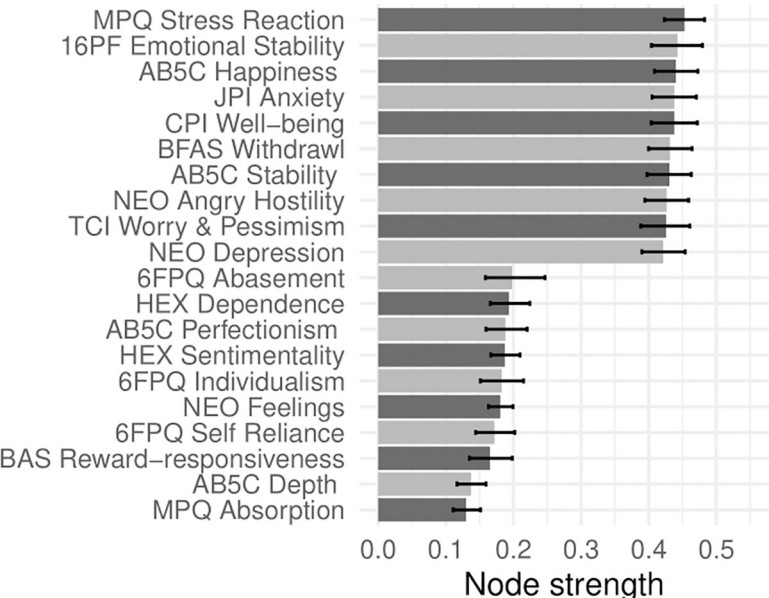

**Fig 9. The most strongly and weakly connected facets in the neuroticism network.**

and conscientiousness. This suggests that, although each aspect scale was derived using items that measured a single domain, some are not factor-pure markers of a single domain [62].

## Discussion

In this study, we created a facet atlas to organize and summarize the current state of facet-level personality trait measurement. To illustrate the utility of this atlas, we examined the prevalence of blended facets, identified core and peripheral facets for each of the Big Five, and explored

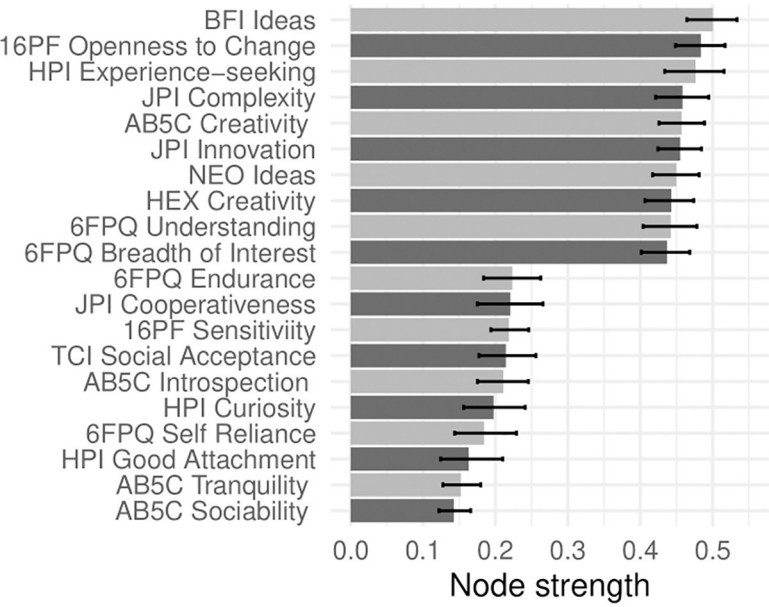

**Fig 10. The most strongly and weakly connected facets in the openness network.**

how researchers can use this facet atlas to better understand particular constructs and measures. In what follows, we discuss implications of this research for applied personality assessment and our conceptual understanding of personality trait structure.

## The prevalence of blended facets

Most scales (59%) contained a blend of multiple Big Five domains. Although we are certainly not the first to point out the lack of simple structure in the personality hierarchy (see [11,15,30]), basic and applied personality researchers seem reluctant to incorporate this complex, blended reality into personality assessment and theory, as evidenced by the simple structure implied in most non-circumplicial personality measures (e.g. [8,10,12]). More fully acknowledging the prevalence of blended content in facet scales can increase commensurability between measures and improve structural theories.

Results also indicated that three specific domain combinations were commonly represented through blends. One of these, the blend between agreeableness and extraversion, reflects the well-known interpersonal circumplex [23]. Two other blends, between agreeableness and neuroticism and between conscientiousness and openness, were also common but have been given less empirical attention. Blends of agreeableness and neuroticism may form a circumplex measuring interpersonal affective tendencies [31]. Conceptualizing such a circumplex may be useful in the diagnosis and treatment of interpersonal problems. The blend between conscientiousness and openness may form a circumplex based around a person's value system (high C high O = ego development, High C low O = rigidity, low C high O = unconventionality, low O low C = disengagement). This circumplex may be useful for understanding humanistic aspects of personality and may aid in synthesizing Big Five personality research with that on ego development (e.g., [63]).

Two particular types of facet blends, between extraversion and openness, and between agreeableness and conscientiousness, were uncommon in this facet atlas. This finding was relatively surprising, as these two pairs of domains are typically intercorrelated [4, 13]. The paucity of blends between facets of extraversion and openness does not appear to reflect empty space, as we did identify a few positive blends (TCI exploratory excitability and AB5C leadership) and a few contrasting blends (AB5C introspection and sociability). Also, past circumplicial research has also found that facets measuring ingenuity, creativity, and bold leadership measure a blend of high openness and high extraversion [30,64]. Rather, it seems that scales measuring blends of extraversion and openness are just uncommon. Developing scales that explicitly measure a blend of these traits may be useful, as they form the metatrait of plasticity and have been theorized to function in tandem as part of the approach system [65]. In comparison, the lack of blended content between agreeableness and conscientiousness may reflect trait space that is necessarily sparser. We found one negative blend (AB5C rationality) and three positive blends (AB5C dutifulness, AB5C morality, and HEXACO fairness), and other research has similarly struggled to identify blended content between these two traits [65]. We note that each of these four facets connotes some sort of interpersonally-focused rule adherence, which is a common behavior in everyday life but does not seem to be well-encoded into the language of stable individual differences (as evidenced by the fact that we cannot identify a single-adjective term to describe this kind of behavior). Future research may wish to further delve into the personality space occupied by a blend of agreeableness and conscientiousness, and to develop scales that explicitly measure this content. Doing so could augur a more comprehensive approach to personality assessment.

## The cores of each domain

We identified the most core and peripheral facets in each Big Five domain by computing each facet's strength within the respective network. A heterogenous set of facets characterized the

five domain cores. For example, the core of conscientiousness contained facets measuring content including mastery, purposefulness, and organization. These findings identify cases of jingle and jangle in facet names. For example, AB5C *sociability* was located in the core of extraversion and JPI *sociability* was in the periphery, though they share the same name. This pattern of findings also highlights the inherent difficulty in identifying a single conceptual "core" for each of the broad Big Five domains. Rather, a domain's core may be best understood as collection of facets, and the positioning of facets may be best done in relative terms (e.g. as "more core" or "more peripheral" than another facet.)

We also found that, within each domain, network centrality estimates were highly correlated with the absolute value of factor loadings in that domain (rs = .71-.86). The magnitude of this correlation, though smaller than the near-unity correlations between these two parameters when estimated in simulation studies [66], suggests that similar information is gleaned from both kinds of analyses (we note that the simulations estimated networks based on partial correlations, whereas we estimated networks based on full correlations). The major source of discrepancy between the two estimates likely comes from the fact that factor analysis summarize how facets are similar in terms of their associations with a single broader Big Five domain, whereas strength centrality estimates summarize all sources of similarity and difference between each pair of facets. For example, TCI social acceptance and AB5C empathy both load strongly on a latent agreeableness factor, but this association is made even stronger by virtue of a shared secondary loading on openness, which is solely captured in network strength estimates. Overall, this overlap between factor analytic and network analytic results suggests that the two methodologies share many features, especially when network analyses are based on cross-sectional correlations.

## The peripheries of each domain

Whereas much attention has been paid to identifying the cores of each of the Big Five domains, this study was one of the first to examine the peripheral facets of the five domains. Results indicated that each domain contained substantial peripheral content that was covered by few measures. Along with past research [34, 36], this suggests that most modern personality measures have idiosyncratic breadth in their content coverage. For example, the NEO-PI-R and HPI measure *trust*, a facet of agreeableness, but the other hierarchical personality measures in the ESCS do not. These differences in peripheral content coverage may account, in part, for the moderate correlations that have been reported for different measures of the same Big Five domain (e.g. as low as *r* = .66 in [10]). Future research that focuses on the periphery of the trait domains can help resolve differences in domain scores from different measures, clarify how different instruments are more or less effective at accounting for certain traits, and improve efforts to comprehensively assess personality.

## Limitations

The major limitations of this research involve the composition of the ESCS. The sample is ethnically homogenous; over 98% of participants are white, all are Americans, and most are middle-aged. As the structure of personality traits, especially at the facet level, does not generalize across cultures [67] or age groups [68], researchers should be cautious when generalizing this atlas to different groups of people. As a broader point, the ESCS has been heavily utilized in past examinations of personality structure (e.g. [10, 12, 30)]) because participants have completed such a wide variety of personality measures. The unfortunate side effect of this overreliance on the ESCS and samples with similar composition is that our research on personality structure often excludes broader nonwhite populations, even within the US. To rectify this,

future work that collects data used to study personality structure using many facet scales must actively focus on sample diversity (such as [69]). We eagerly anticipate future, more representative atlases.

In addition, all of the instruments in this study were self-report questionnaires, and results may differ when using a different method. Additionally, some facet scales were measured with few items, and this brevity introduces measurement unreliability. We corrected for this using each scale's alpha reliability, but this rough correction is relatively conservative and may not restore each correlation to its actual magnitude. As such, correlations between facets measured with brief scales may be somewhat attenuated.

## Conclusion

We created a facet atlas that organizes the current state of facet-level measurement by describing connections between different lower-order personality trait scales. A better understanding of facets can assist in individual case formulation [28], clarify trends in personality development [70], aid in the prediction of important life outcomes [71], and refine theories of personality structure [72]. Ideally, this atlas can serve as a reference guide to personality scholars of all stripes who wish to use facet scales in research and applied settings.

## Author Contributions

**Conceptualization:** Ted Schwaba, Christopher J. Hopwood, Wiebke Bleidorn.

**Formal analysis:** Ted Schwaba, Mijke Rhemtulla.

**Methodology:** Mijke Rhemtulla, Christopher J. Hopwood, Wiebke Bleidorn.

**Software:** Ted Schwaba.

**Supervision:** Wiebke Bleidorn.

**Visualization:** Ted Schwaba.

**Writing – original draft:** Ted Schwaba, Christopher J. Hopwood, Wiebke Bleidorn.

**Writing – review & editing:** Ted Schwaba, Mijke Rhemtulla, Christopher J. Hopwood.

**Writing - Review & Editing:** Wiebke Bleidorn.

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
