## [Decision Letter · Decision Letter 0]

17 Jun 2020

PONE-D-20-09874

The facet atlas: Using network analysis to describe the blends, cores, and peripheries of personality structure

PLOS ONE

Dear Dr. Schwaba,

Thank you for submitting your manuscript to PLOS ONE. After careful consideration, we feel that it has merit but does not fully meet PLOS ONE’s publication criteria as it currently stands. Therefore, we invite you to submit a revised version of the manuscript that addresses the points raised during the review process.

We look forward to receiving your revised manuscript.

Kind regards,

Frantisek Sudzina

Academic Editor

PLOS ONE

Journal Requirements:

2. Please upload a copy of Figure 1-5, to which you refer in your text on page 19. If the figure is no longer to be included as part of the submission please remove all reference to it within the text.

Additional Editor Comments (if provided):

Please take time to think which suggestions when implemented in what why will improve the article.

Reviewers' comments:

Reviewer's Responses to Questions

**Comments to the Author**

1. Is the manuscript technically sound, and do the data support the conclusions?

Reviewer #1: Yes

Reviewer #2: Yes

Reviewer #3: Partly

Reviewer #4: Partly

2. Has the statistical analysis been performed appropriately and rigorously? 

Reviewer #1: Yes

Reviewer #2: Yes

Reviewer #3: Yes

Reviewer #4: Yes

3. Have the authors made all data underlying the findings in their manuscript fully available?

Reviewer #1: Yes

Reviewer #2: Yes

Reviewer #3: Yes

Reviewer #4: Yes

4. Is the manuscript presented in an intelligible fashion and written in standard English?

Reviewer #1: Yes

Reviewer #2: Yes

Reviewer #3: Yes

Reviewer #4: Yes

5. Review Comments to the Author

Reviewer #1: The authors present a practical tool for personality psychology and a concise study justifying it. Their walkthrough of exploring impulsivity facets was especially helpful in showcasing the utility of their online app. I have one primary concern and two requests for additional information to be added to supplemental materials.

1. My biggest concern is that node strength is not an especially informative metric. As far as I have been able to discern from Figures 6-10, for a given domain, a facet’s strength only indicates whether it falls into one of two categories: strong or weak. This information is better than nothing. To give the figure more context, it would be helpful to at least add to each figure the maximum allowable strength for facets within the domain. It appears as if between-domain facets differ in their strength (e.g., the strongest Neuroticism facets are stronger than the strongest Conscientiousness facets), but upon closer inspection (which a non-reviewer may not bother with), it is apparent that these differences are due to the number of facets in each domain (e.g., 107 facets in Neuroticism [and thus a max strength of 106?], vs. 70 facets in Conscientiousness [and thus a max strength of 69?])

Even with the above modification, node strength is not an effect size that is interpretable in and of itself, nor is it comparable across domains or studies; a new, similar study will mostly likely use a different number of within-domain facets than this current one. You should strongly consider an alternative (or additional) metric to node strength—the average of absolute correlations, as opposed to the sum. An average correlation may be found by converting each correlation into a Z value, taking the mean of the Z values for a given facet, and back-transforming the mean Z value into a correlation (see Corey, Dunlap and Burke, 1998 as to why this method is preferred over simply taking the linear mean of correlations). Within a domain, the order of strongest-to-weakest facets should remain largely the same, if not identical, for the average vs. the sum of absolute correlations. However, an average correlation will be an interpretable effect size that is comparable across domains and studies. You should be able to find the confidence intervals of the correlations with a similar bootstrapping method that you used for your node strength intervals.

2. I am sympathetic to the fact that a full correlation matrix would take 30 sheets of paper to display, and I agree that it’s not appropriate to show in the manuscript, but a full matrix would only require one .CSV file in the supplemental materials. Having the full correlation matrix (raw below the diagonal, corrected for attenuation above, as in Table 3) is useful for transparency and replication. For example, without the full correlation matrix, a reader would not be able to replicate your analysis of the four impulsivity facets; they would not be able to reconstruct Table 3. Also, they would not be able to construct a new correlation table of supposedly-similar facets. For performing future analyses, a full correlation matrix is probably as critical as the facet atlas. Please add it (e.g., a .CSV or R data file) to the supplemental materials. I don’t think you need to go to the effort of marking whether each correlation is significant (in fact, it’s better if the matrix doesn’t contain anything besides variable names and correlation values, so it can be easily read by programs like R). You could just make a general statement about significance (e.g., to be conservative, use the minimum N to determine the smallest absolute correlation that would be statistically significant (p<.05).

3.“Before visualizing the facet atlas, we first organized all 268 ESCS facets into the Big Five domains using exploratory factor analysis with oblimin rotation” (p. 10). So you created an aggregated Big Five on which to load all facets. I am curious how, for a given aggregated domain, different measures of that domain load onto it. It is probably the case, for example, that there is variability in how strongly different measures of Conscientiousness load onto your aggregated Conscientiousness factor. This information would also be useful to researchers, and may provide context as to why a facet is peripheral (e.g., its domain is also peripheral to other domain measures or has low loadings on the corresponding aggregated domain), although I acknowledge that this is a bit of a tangent from your paper, so it is my least pressing request for supplemental material.

4. “All facets with factor loadings greater than |.25| were included in that domain.” Just to double-check, were there any facets that did not load onto any of the five domains? If so, these facets should be named.

5. There were a number of punctuation typos scattered throughout the manuscript (e.g., on page 17: “Researchers interested in studying impulsiveness may benefit from considering these components separately or in tandem [60]. and from paying close attention to the scales used in past research on impulsiveness”). The draft was well-written, but you should re-read the draft carefully to find these mistakes.

Reviewer #2: I had mixed reactions to this manuscript. On the whole, I think this is a very useful contribution, and the atlas may be very useful for the personality field. However, I had some methodological questions as I went through it, and I struggled to make the online version of the atlas work in any stable way.

One of the questions I had, was why the title and introduction were framed by network analyses, when so much of the methods were based on factor analysis. To me it seemed like the study couldn't be completed without the factor analyses, and indeed much of the findings rested on decisions for the factor analysis. To me it seems like this atlas was at least as strongly based on factor analytic findings as it was on network findings. So, I'm not sure I think the title accurately reflects what the methods the manuscript relies on.

Another question I had was how much the network incremented or gave different results to what might be gleaned from the factor analytic findings. In other words, is there anything we are learning beyond what we could have learned by looking at the factor loadings in the exploratory factor analysis that the authors ran to decide what facets to include in the network analyses? Given how overlapping these methods are, is one just a visual depiction of the results of the other? I'd really like to know if there are any meaningful differences between the strength centrality estimates for one domain and the factor loadings for the same domain. In simulation work cited by the authors (Hallquist et al., in press), the correlation between factor loadings and strength centrality is > .90. And, if there are any meaningful differences why do the authors think those come about and what is their meaning? For instance, are they perhaps due to secondary factors that are not explicitly modeled in the network analysis? Like, are centrality estimates meaningfully impacted by how much secondary content is in a network in a domain? Or how many scales from a particular measure are included? Simulations suggest that additional shared data generating mechanisms (e.g., methods, secondary content) will increase strength centrality estimates. Maybe these are minor relative to the full domain, but that would be interesting to know. A virtue of factor analysis is the ability to partition variance explicitly, but that seems to be something that network analysis can't do or struggles to do.

I don't know the answer to these questions vis-a-vis these particular analyses, but I found myself wondering about them as I read the paper. I guess one direct implication is that the authors might want to do a better job of highlighting what the unique contribution of each method is to arriving at these conclusions.

I also struggled with the user interface for the online resources. To be clear, I think this is a very useful and exciting contribution, but it wasn't clear to me how to use it, and without the manuscript it wouldn't be useful as a stand alone app. So I think they authors should spend some time on providing documentation there. Additionally, I was kicked off the server basically every time I tried to change a parameter and play around with it.

In the intro I thought there were some places that presented somewhat caricatured descriptions of factor and network analysis. At other times, in the discussion, it would seem that some of the statements seemed too strong (e.g., like how the network provides a 'stringent test' of whether aspect scales are maximally distinct, but then this was followed by what sounded like an impressionistic statement of distance looking at the graph).

In sum, I think providing this sort of summary atlas of the structure of many different personality scales is a very useful contribution to the field. I think there is some room in clarifying what the methods used in the study are incrementally adding to each other, but this should be relatively easy for the authors to address in a revision. In particular, I think quite a few readers will likely be interested to know whether there are discrepancies between the findings used across methods, or whether the networks are just visually depicting what the factor analyses already revealed. I hope these comments are helpful in strengthening the presentation of this interesting work.

Sincerely,

Aidan Wright, PhD

Associate Professor

University of Pittsburgh

Reviewer #3: I'm glad for the chance to review this work. The manuscript is very well done, and I think it stands to make an important contribution to measurement in personality psychology. This is not trivial, in my opinion, for measurement issues are closely tied to one of the central missions of personality: to defensibly account for the full breadth of psychological individual differences. In fact, I think this project has already succeeded in this respect, as many researchers working in this area are already familiar with it (and like it!).

This last comment is somewhat unusual, coming from a reviewer, so I want to begin by detailing my familiarity and knowledge of this paper. I hope the editor and the authors will take this prior knowledge into account when weighing my review. I also think it means the authors would benefit from feedback from multiple other reviewers, if at all possible. I reviewed this manuscript for another journal last year. Before that I was made aware of the proposed work due to my involvement with a special issue at another journal. Separate from these experiences, two other researchers have contacted me to ask my opinion of this work since submitting my last review. Before accepting this current review assignment, I sent an inquiry to the action editor regarding the possibility of posting an open review alongside the manuscript. I did not hear back, but I would be interested in that option if all parties are amenable.

I think this work should be published because it uses novel statistical methods and a unique, large, publicly-available dataset. I also think it could be made better with some revision. It does not appear that the manuscript has been altered since my last review, at least not substantially, so I will attach that review to this one for the authors' consideration. I retract point 4 from that review because I now see that it is well-addressed in the main text (previously in a footnote I had overlooked). Of course, I do not think all the points in my review need to be incorporated into the manuscript.

There is one point however that I think must be addressed, and this is the reason why I accepted this review assignment. The sample is 98.4% white. These are basically all white middle-aged homeowners from a small rural area in Oregon. The sample has been used widely among personality psychologists over the last 20 years in order to develop public-domain alternatives of proprietary personality scales. But, there is no solid data showing that the structure of personality based on these data is generalizable. I understand the rationale for calling this tool "The Facet Atlas" based on analogous work(s) from genetics, but the analogy is far from perfect. To suggest that this is *the* structure of facets is more than just a theoretical discussion about the required scope of cross-validation. I believe it has real, detrimental consequences for the field.

I don't feel it would reduce the impact of the manuscript to make this issue a central theme. Currently, it is addressed in passing on line 170 and in more detail on lines 420 to 423. As a digression, please note on lines 422-423 that it is, sadly, a dramatic understatement to say that "the structure of personality does not generalize perfectly across cultures." Not even the Big Five generalizes (see de Raad et al., 2010 and the large related literature on this issue); there is little expectation that the facet- or item-level structure generalizes even moderately across cultures. But that is not a failing of this paper! It is the expected reality of phrased personality items (containing contextualized content) when administered to a diverse range of cultures and sub-cultures.

As I write this review, protestors across the U.S. and beyond are rallying in condemnation of pervasive, systemic racism. My field (personality), my institution, and even my own research program contribute to this persistently, if only on the margins and without intention. One way that diversity and inclusivity can be addressed is to re-evaluate whether the consequences of our claims do more harm than good. I would be remiss if I did not point out (again) that this paper is an opportunity to describe these widely-used data as deeply flawed, and then frame the work as a methodological advancement, ready for application to additional data sets, if/when they become available in the future. And maybe to also call for the urgent need to collect such data! Several research teams are working on this.

Regards,

David Condon

Reviewer #4: I’ll note my identity at the outset as Doug Samuel as this is relevant in terms of possible Conflicts of Interest. Primarily in that I have collaborated with some of these authors recently. I have also worked with Dr. Hopwood on numerous projects in the past and consider him a friend. I also note that my graduate student (Meredith Bucher) and I have been doing somewhat related work in terms of trying to organize facet scales. We have done so using the AB5C framework explicitly (Bucher & Samuel, 2018 in Journal of Personality Assessment; and other Bucher & Samuel that was just accepted pending revisions at Journal of Research in Personality). I don’t see either of our papers as competing with the present effort, but do want to be upfront about an possible COIs as I suggest references below.

This is a paper that uses the Eugene Springfield Community Sample (ESCS) to conduct a network analysis of 268 lower-order scales, across 13 personality measures to determine how closely the relate to each other and arrive at an estimate of which content was most common across the different measures. This led to an interesting product of a web-based app that would allow this information to be adapted and used by researchers or clinicians. I found this paper to be clear and easy to read. It had a number of older citations that showed a good command of seminal literature. That said, the authors unfortunately missed some very relevant citations that are key to this paper, including a major one that undertook very similar analyses in this same dataset. Woods and Anderson (2016) also used the ESCS to examine the degree of overlap of facet scales with the ultimate goal of creating a “periodic table of personality” (Woods, S. A., & Anderson, N. R. (2016). Toward a periodic table of personality: Mapping personality scales between the Five-Factor Model and the circumplex model. Journal of Applied Psychology, 101, 582–604.). This Woods and Anderson paper covers much of the same ground as the present paper and as such it simply must be incorporated into the background. My opinion is that there is sufficient differences in method, as well as product that it does not suggest this current paper shouldn’t be published, but it will need to be contextualized in terms of what has already been done and compared to those results. I think this becomes most tricky in terms of the wording. That paper aimed to provide a “periodic table of personality” and this one aims for a “personality atlas.” IMO, there may be differences there, but they are subtle and the present authors will need to take great care to explain how this is different and/or the same.

Additionally, the authors should consider citing a 2016 editorial by Ziegler & Backstrom that appeared in European Journal of Psychological Assessment (issue 32, pages 105-110) that also took a look at trying to sort through various facets. Another work to consider that is aimed at a similar goal is by David Condon and Bill Revelle. I am honestly not sure what is the best citation for it, but this one might be it (https://openpsychologydata.metajnl.com/articles/10.5334/jopd.32/). My understanding was that they were aiming to create an omnibus measure that represented a consensus list of facets across many inventories.

As I mentioned, Meredith Bucher and I have published one paper that created a short form of the AB5C and then used new data to sort major faceted inventories (HEXACO and NEO) into the that context. The AB5C specifically is very relevant to the present paper as the authors repeatedly focus on the circumplex tradition and the promise it holds for understanding facets (page 4 of the present paper)…which is of course the entire idea behind the AB5C, so it would be helpful to cite as support here. In our 2018 paper (JPA; issue 101:1, 16-24) we found that some domain blends were less well-represented than others, so this is quite relevant to current discussion. In particular, our 2nd paper is specifically looked at those facets of the AB5C that weren’t performing quite as expected in terms of primary and secondary loadings to see if new items (from the IPIP) would improve this. Interestingly, we find that the specific facet that blends high agreeableness and low conscientiousness was not able to be measured well. We interpret this as being an area of personality space that may well be less populated. We don’t go so far as to say its vacant, but the combination of findings across studies suggest there are few, if any, words in the lexicon that map into this space and we couldn’t not locate suitable items to measure that combination. In short, it seems to be an area of the personality space that seems to have been uncommon (or unimportant) that it was not encoded into language or common measures. We hope to have this accepted soon, but I’d be happy to share the version informally. As it pertains here, I’d appreciate the authors digging more into those less populated facets in their writing.

As I noted above, the creation of the app was really a nice feature of this paper and I do think it has the possibility to be useful to the field. That, combined with the network analyses are likely sufficiently novel to be a meaningful extension to the literature. The authors would do well to say more about specifically how the network models are different from what was reported in Woods & Anderson (2016).

The authors do a reasonable job of noting the strengths and weaknesses of the ESCS. The demographics are less than ideal, but it has a great set of measures (that is thorough, but not comprehensive). That said, more needed to be said about the complications of correlating measures that were administered up to a decade apart from each other.

This is important as it relies upon the input to dictate the output. These are more “consensus” than completeness (e.g., it won’t “resolve” anything”). A key limitation is that the “items in some scales were administered over a period of multiple years” as noted on page 9. Some scales you are correlating are measured a decade apart. What impact might this have? It was also not clear how big the sample was for the network analyses. It is not clear, but I would understand based on the variability in Table 3 that some sort of deletion was used for given correlations. But when this was for the entire collection of measures, did you also use listwise deletion? If so what was the final sample size? Or were missing data imputed?

I also encourage the authors to clearly define what they mean by “core.” Given the method, it means here that is the most common within the measures used here. It is NOT the lexical core or mean to be the “center” of the domain. I personally would prefer a term like “consensus” or the “common denominator” as this more clearly reflects what it means. This has important implications for the discussion in particular as the authors wrote “Network theory suggests that change to a network’s central nodes may cascade through the rest of the network [32]. This idea is appealing to intervention researchers who seek to use focal interventions to enact relatively far-reaching personality change [66]. Figures 1-5 provide some guidance about which facets are more or less core, and thus may represent better or worse targets for domain wide change efforts.” IMO, this is not necessarily correct. These analyses tell us what is most commonly included among these 13 measures. That doesn’t mean they are at the center of the lexical factor, nor that it is causally core in the way that is implied by the above example. We would need more data to make that comparison; ideally longitudinal data.

Similarly, what you call periphery doesn’t mean it is “less relevant” (although it might), but here it only means it is less common across these measures. For all we know, a very uncommonly assessed facet might turn out to the be the most “central” or “core” of a domain.

Minor points:

1. I was not certain of the point of the BFAS-specific focus in the discussion. Why single out the BFAS over any of the other measures?

2. I was surprised to see the authors correct using Cronbach’s alpha. Of course, as the authors know, alpha is not an index of true reliability. Les Morey’s address at SPA in San Francisco a few years ago provided a number of examples of how alpha was not a measure of reliability and why correcting using that could lead to skewed estimates of this. Test-retest dependability would be a much stronger correction. In the absence of test-retest over a short period (Watson, 2004), I would encourage the authors to just abandon corrections at all. I realize that may sound extreme so I’d also be content if they simply provided the uncorrected results as well.

3. On page ?? you say with regard to blended traits that “basic and applied personality researchers seem reluctant to incorporate this complex, blended reality into personality assessment and theory.” Can you give an example of the relucatance? I don’t personally see such a reluctance, but maybe I’m not clear on what is being communicated.

6. PLOS authors have the option to publish the peer review history of their article (what does this mean?). If published, this will include your full peer review and any attached files.

Reviewer #1: Yes: Lorien G. Elleman

Reviewer #2: No

Reviewer #3: Yes: David M Condon

Reviewer #4: Yes: Douglas B. Samuel

---

## [Author Response · Author response to Decision Letter 0]

13 Jul 2020

***

To: 

Dr. M. Frantisek Sudzina

Academic Editor

PLOS ONE

***

Dear Dr. Sudzina,

Please find below our responses to the requested revisions.

Sincerely,

Ted Schwaba, Mijke Rhemtulla, Chris Hopwood, and Wiebke Bleidorn

***

Additional Editor Comments (if provided):

Please take time to think which suggestions when implemented in what why will improve the article.

We have responded to all reviewer comments below, and have integrated their suggestions into the manuscript.

***

Reviewer #1: 

1. My biggest concern is that node strength is not an especially informative metric. As far as I have been able to discern from Figures 6-10, for a given domain, a facet’s strength only indicates whether it falls into one of two categories: strong or weak. This information is better than nothing. To give the figure more context, it would be helpful to at least add to each figure the maximum allowable strength for facets within the domain. It appears as if between-domain facets differ in their strength (e.g., the strongest Neuroticism facets are stronger than the strongest Conscientiousness facets), but upon closer inspection (which a non-reviewer may not bother with), it is apparent that these differences are due to the number of facets in each domain (e.g., 107 facets in Neuroticism [and thus a max strength of 106?], vs. 70 facets in Conscientiousness [and thus a max strength of 69?]).

 Even with the above modification, node strength is not an effect size that is interpretable in and of itself, nor is it comparable across domains or studies; a new, similar study will mostly likely use a different number of within-domain facets than this current one. You should strongly consider an alternative (or additional) metric to node strength—the average of absolute correlations, as opposed to the sum. An average correlation may be found by converting each correlation into a Z value, taking the mean of the Z values for a given facet, and back-transforming the mean Z value into a correlation (see Corey, Dunlap and Burke, 1998 as to why this method is preferred over simply taking the linear mean of correlations). Within a domain, the order of strongest-to-weakest facets should remain largely the same, if not identical, for the average vs. the sum of absolute correlations. However, an average correlation will be an interpretable effect size that is comparable across domains and studies. You should be able to find the confidence intervals of the correlations with a similar bootstrapping method that you used for your node strength intervals.

We agree that computing strength estimates based on the average correlation, rather than the sum of correlations, is more informative. We have updated Figures 6-10 and the supplemental materials to display strength estimates based on this metric. On p. 12, we now write, 

“In this study we calculate strength as the average of all correlations so that this metric is comparable across Big Five domains and with other networks.” 

2 I am sympathetic to the fact that a full correlation matrix would take 30 sheets of paper to display, and I agree that it’s not appropriate to show in the manuscript, but a full matrix would only require one .CSV file in the supplemental materials. Having the full correlation matrix (raw below the diagonal, corrected for attenuation above, as in Table 3) is useful for transparency and replication. For example, without the full correlation matrix, a reader would not be able to replicate your analysis of the four impulsivity facets; they would not be able to reconstruct Table 3. Also, they would not be able to construct a new correlation table of supposedly-similar facets. For performing future analyses, a full correlation matrix is probably as critical as the facet atlas. Please add it (e.g., a .CSV or R data file) to the supplemental materials. I don’t think you need to go to the effort of marking whether each correlation is significant (in fact, it’s better if the matrix doesn’t contain anything besides variable names and correlation values, so it can be easily read by programs like R). You could just make a general statement about significance (e.g., to be conservative, use the minimum N to determine the smallest absolute correlation that would be statistically significant (p<.05).

On p. 10, we now write, 

“A correlation matrix of all facets is available at https://osf.io/w682t/”

3. “Before visualizing the facet atlas, we first organized all 268 ESCS facets into the Big Five domains using exploratory factor analysis with oblimin rotation” (p. 10). So you created an aggregated Big Five on which to load all facets. I am curious how, for a given aggregated domain, different measures of that domain load onto it. It is probably the case, for example, that there is variability in how strongly different measures of Conscientiousness load onto your aggregated Conscientiousness factor. This information would also be useful to researchers, and may provide context as to why a facet is peripheral (e.g., its domain is also peripheral to other domain measures or has low loadings on the corresponding aggregated domain), although I acknowledge that this is a bit of a tangent from your paper, so it is my least pressing request for supplemental material.

In Figures 1-5, we visualize the Big Five factor loadings for each facet, and these loadings are presented numerically in the supplementary descriptives table. These factor loadings describe how strongly each scale loads onto each factor-analytically derived domain. Readers can use this information to connect each Big Five facet scale to its intended domain and compare these loadings across facets.

4. “All facets with factor loadings greater than |.25| were included in that domain.” Just to double-check, were there any facets that did not load onto any of the five domains? If so, these facets should be named.

There were. On p. 11, we now write, 

“Two facets, HPI Not Autonomous and HPI Not Spontaneous, did not load .25 on any domain and thus were not included in any network.”

5. There were a number of punctuation typos scattered throughout the manuscript (e.g., on page 17: “Researchers interested in studying impulsiveness may benefit from considering these components separately or in tandem [60]. and from paying close attention to the scales used in past research on impulsiveness”). The draft was well-written, but you should re-read the draft carefully to find these mistakes.

To address this comment, we have gone through and proofread the manuscript an additional time, correcting this typo and others. 

***

Reviewer #2: 

1. One of the questions I had, was why the title and introduction were framed by network analyses, when so much of the methods were based on factor analysis. To me it seemed like the study couldn't be completed without the factor analyses, and indeed much of the findings rested on decisions for the factor analysis. To me it seems like this atlas was at least as strongly based on factor analytic findings as it was on network findings. So, I'm not sure I think the title accurately reflects what the methods the manuscript relies on.

We agree that the factor analysis in this study was a critically important intermediate step, as it allowed us to empirically categorize each facet into a Big Five network and provided us with information about facet blendedness. However, we believe the main contribution of the present manuscript comes from the final network visualizations. To more accurately communicate that the networks were not solely derived using network analytic methods, we have replaced the title phrase “using network analysis to describe the blends, cores, and peripheries of personality structure” with “visualizing networks that describe the blends, cores, and peripheries of personality structure”

2. Another question I had was how much the network incremented or gave different results to what might be gleaned from the factor analytic findings. In other words, is there anything we are learning beyond what we could have learned by looking at the factor loadings in the exploratory factor analysis that the authors ran to decide what facets to include in the network analyses? Given how overlapping these methods are, is one just a visual depiction of the results of the other? I'd really like to know if there are any meaningful differences between the strength centrality estimates for one domain and the factor loadings for the same domain. In simulation work cited by the authors (Hallquist et al., in press), the correlation between factor loadings and strength centrality is > .90. And, if there are any meaningful differences why do the authors think those come about and what is their meaning? For instance, are they perhaps due to secondary factors that are not explicitly modeled in the network analysis? Like, are centrality estimates meaningfully impacted by how much secondary content is in a network in a domain? Or how many scales from a particular measure are included? Simulations suggest that additional shared data generating mechanisms (e.g., methods, secondary content) will increase strength centrality estimates. Maybe these are minor relative to the full domain, but that would be interesting to know. A virtue of factor analysis is the ability to partition variance explicitly, but that seems to be something that network analysis can't do or struggles to do.

To address this comment, we correlated the strength centrality estimates of each network with the factor loadings of the facets in each network. On p. 15, we now write, 

“Strength centrality estimates were highly correlated with the absolute value of factor loadings on that domain. Correlations were .82 for extraversion, .76 for agreeableness, .86 for conscientiousness, .79 for neuroticism, and .71 for openness.”

On p. 21, we discuss the implications of these results: 

“[W]ithin each domain, network centrality estimates were highly correlated with the absolute value of factor loadings in that domain (rs = .71-.86). The magnitude of this correlation, though smaller than the near-unity correlations between these two parameters when estimated in simulation studies [35], suggests that similar information is gleaned from both kinds of analyses (we note that the simulations estimated networks based on partial correlations, whereas we estimated networks based on full correlations). The major source of discrepancy between the two estimates likely comes from the fact that factor analysis summarize how facets are similar in terms of their associations with a single broader Big Five domain, whereas strength centrality estimates summarize all sources of similarity and difference between each pair of facets. For example, TCI social acceptance and AB5C empathy both load strongly on a latent agreeableness factor, but this association is made even stronger by virtue of a shared secondary loading on openness, which is solely captured in network strength estimates. Overall, this overlap between factor analytic and network analytic results suggests that the two methodologies share many features, especially when network analyses are based on cross-sectional correlations.”

3. I also struggled with the user interface for the online resources. To be clear, I think this is a very useful and exciting contribution, but it wasn't clear to me how to use it, and without the manuscript it wouldn't be useful as a stand alone app. So I think they authors should spend some time on providing documentation there. Additionally, I was kicked off the server basically every time I tried to change a parameter and play around with it.

We have markedly improved the UI of the app. To specifically address this comment, we have added an “info” tab that provides background information geared to both “Network noobs” (a plain language summary of how the network works) and “Network vets” (technical details akin to those found in the methods section), allowing the app to better stand on its own. We have also updated the R packages in the app to their latest versions, leading to fewer server disconnects (although disconnects sometimes still occur when multiple users use the app at once, which is unfortunately a drawback of shiny apps in general). As a workaround, users can now download and run an offline version of the app, and instructions for how to do this are on the info tab.

Additionally, we have expanded the descriptive information provided about each facet, which is now included in a tab labeled “descriptives,” and we have reworked the UI of the main page so that the networks look prettier and it is easier for users to connect facet numbers to names. 

4. In the intro I thought there were some places that presented somewhat caricatured descriptions of factor and network analysis. 

We have revised these instances to more accurately summarize the similarities and differences between these two methods while retaining the manuscript’s readability to a more general audience. For example, on p. 8, we now summarize the two methods as follows,

 “Network graphs also provide a complementary approach to factor analyses. Factor analysis allows researchers to distill a large covariance matrix into a smaller factor structure that captures the main dimensions of variability among items, such as the Big Five. Conversely, network graphs retain focus on the lower-level variables (in this case facets). As such, network visualizations depict individual variable-to-variable correlations, which are not the focus of factor analytic results.”

5. At other times, in the discussion, it would seem that some of the statements seemed too strong (e.g., like how the network provides a 'stringent test' of whether aspect scales are maximally distinct, but then this was followed by what sounded like an impressionistic statement of distance looking at the graph).

We have revised the discussion section so that the strength of our statements are better calibrated to our tests of those statements. Regarding the reviewer’s specific example, we now write on p. 18 that

“A consequence of the specific factor-analytic procedure used to derive the BFAS is that, because the two aspect scales in each domain were rotated to be orthogonal, they should each measure distinctly different content. This can be evaluated by examining how far apart the two aspects are within each domain’s network, as scales that have similar patterns of correlations with other facets will occupy network positions close to one another. Results indicated that the two BFAS aspect scales in each domain were indeed placed relatively far apart from one another (while still remaining in the domain’s core), supporting the idea that these scales cover a relatively broad content area within each domain.” 

***

Reviewer #3: 

1. There is one point however that I think must be addressed, and this is the reason why I accepted this review assignment. The sample is 98.4% white. These are basically all white middle-aged homeowners from a small rural area in Oregon. The sample has been used widely among personality psychologists over the last 20 years in order to develop public-domain alternatives of proprietary personality scales. But, there is no solid data showing that the structure of personality based on these data is generalizable. I understand the rationale for calling this tool "The Facet Atlas" based on analogous work(s) from genetics, but the analogy is far from perfect. To suggest that this is *the* structure of facets is more than just a theoretical discussion about the required scope of cross-validation. I believe it has real, detrimental consequences for the field.

I don't feel it would reduce the impact of the manuscript to make this issue a central theme. Currently, it is addressed in passing on line 170 and in more detail on lines 420 to 423. As a digression, please note on lines 422-423 that it is, sadly, a dramatic understatement to say that "the structure of personality does not generalize perfectly across cultures." Not even the Big Five generalizes (see de Raad et al., 2010 and the large related literature on this issue); there is little expectation that the facet- or item-level structure generalizes even moderately across cultures. But that is not a failing of this paper! It is the expected reality of phrased personality items (containing contextualized content) when administered to a diverse range of cultures and sub-cultures.

As I write this review, protestors across the U.S. and beyond are rallying in condemnation of pervasive, systemic racism. My field (personality), my institution, and even my own research program contribute to this persistently, if only on the margins and without intention. One way that diversity and inclusivity can be addressed is to re-evaluate whether the consequences of our claims do more harm than good. I would be remiss if I did not point out (again) that this paper is an opportunity to describe these widely-used data as deeply flawed, and then frame the work as a methodological advancement, ready for application to additional data sets, if/when they become available in the future. And maybe to also call for the urgent need to collect such data! Several research teams are working on this.

We agree that this is an important, and timely, issue. To address concerns about generality of these findings, we have retitled the paper “A facet atlas” rather than “The facet atlas” and we have revised the text in the manuscript accordingly. We have also restructured the paragraph about limitations of the ESCS to better communicate limitations of this dataset. On p. 22, we now write, 

“As a broader point, the ESCS has been heavily utilized in past examinations of personality structure (e.g. (10,12,30)) because participants have completed such a wide variety of personality measures. The unfortunate side effect of this overreliance on the ESCS and samples with similar composition is that our research on personality structure often excludes broader nonwhite populations, even within the US. To rectify this, future work that collects data used to study personality structure using many facet scales must actively focus on sample diversity (such as (71)). We eagerly anticipate future, more representative atlases.”

2. I would not call this "The Facet Atlas", for two reasons. First, many of these frameworks are not measuring facets and this is not a trivial methodological or semantic distinction, no matter how it may seem. The word "facets" has a specific meaning and many of the scales included here are not facets. The BFAS for example (introduced in a paper titled, "Between Facets and Domains: 10 Aspects of the Big Five") and also the MPQ, the 16 PF, and the BIS/BAS scales. (As a side note, it would be nice to have some explanation as to why these frameworks were included as they are not Big Five/FFM based, are not facets, and several of them appear quite peripheral). Second, "The Facet Atlas" suggests a more robust reference than can be justified given the data. Even if the ESCS sample did not suffer from severe homogeneity, I would still want to have multiple samples before using such a term. Maybe "a facet network" or something similar?

We respectfully disagree with the claim that there is a specific, agreed-upon meaning for facets. For example, combing through DeYoung’s work on aspects, that they are not termed facets seems to be an arbitrary decision – they do not differ from other facet solutions in a substantive theoretical way, there are just fewer of them. Other work similarly seems to define facets more by what they aren’t (domains or specific behaviors) rather than by what they are, and we follow that trend here:

 On p. 3, we now clarify our definition of facets:

 “(for the purposes of this paper, we define a personality facet as any personality trait that is narrower than a domain yet broader than a specific behavioral nuance).”

Regarding scale inclusion, on p. 9, we better clarify that

“[We] included omnibus scales with either explicit hierarchical structure, hierarchical structure validated in other studies, or that measure many traits, such that each trait would approximate the scope of a facet.” 

We believe that erring on the side of inclusivity allows this atlas to be applicable to a greater body of existing and future research. 

From attached review provided by reviewer 3:

3. I am not able to access the supplementary materials here, but the description of the sample in the Methods section really should be further developed. The authors reference the summary by Saucier but readers of this article need details -- more educational attainment info (just says 20% have college degrees), more info about the distribution of age (the range suggests that the data are more evenly distributed than is the case), more details about the actual sample sizes (N=1142 is also misleading; what matters is the range of effective n's). I also think it would help to include some description of how the participants were recruited and retained -- they were solicited through a widespread mailer campaign to >10,000 homeowners in the Eugene-Springfield area and were paid (how much? I think $30) to respond to each of the many mailers (how many were needed to collect these data?). These issues of recruitment/retention underscore the likelihood of strong selection effects based on psychological attributes in addition to the obvious concerns about demographics.

We have added more information about ESCS sample composition based on the details provided in the most recent technical report. On p. 9, we now write,

“ESCS participants were recruited through a mailed invitation in 1993 and completed a variety of personality questionnaires sent in separate mailers beginning in 1993 (N = 1,134), with 88% retention over the following 10 years. In 1993, participants ranged in age from 18-85 (M = 49.67, SD = 13.08), and 34.7% of participants were ages 40-49. The sample was composed of 57% women, 98.4% European-Americans, and 59% of participants had at least a college degree. The ethnic and geographical homogeneity of the ESCS limit the generalizability of results, as we note in the discussion section. The number of participants varies by scale, as questionnaires were over a period of 2 decades. The items in some scales were administered over a period of multiple years, leading some facet scales within the same questionnaire to have different sample sizes.”

Additionally, In the abstract and methods section, we now clarify that pairwise Ns = 571-948.

4. Some discussion of the decision to omit the "ability" facets of the HPI should be included in the text of the manuscript. There is debate in the literature about the extent to which the Big Five covers cognitive abilities (see DeYoung, 2011). This seems like a missed opportunity. If these scales are omitted based on evidence that they don't fit (presumably bc they are too peripheral to the Openness network), this is worth detailing.

On p. 10, we write that 

“we omitted nine HPI facets (science ability, intellectual games, education, math ability, good memory, reading, self-focus, impression management, appearance) that measured ability, as these scales did not correlate highly with other facet scales.”

 In other words, these scales did not load highly on any of the Big Five and were extremely peripheral (to the extent that they distorted network visualizations). We believe that this is certainly interesting, but the debate between personality traits and ability is a complex one that goes beyond the scope of this manuscript. 

5. I was generally confused by the section about the BFAS in the Results (currently pages 15 and 16). No rationale/introduction for this work (or the Impulsivity work) is given in advance and I don't see the justification for singling out the evaluation of this measure over some of the others. If anything, I think this serves to reify the BFAS in a manner that is misleading -- as the authors state, the BFAS was developed using the exact same data and measures used here (the AB5C and NEO-PI-R).

If the authors choose to retain this section (which I do not recommend), more support should be given to their claim that new information is gained from these analyses. With respect to the first point, I am confused as to how the aspects could not be in the "core" of each domain? After all, they are derivative of the 75 AB5C and NEO facets. This is not a test of the BFAS structure/orientation at all really but rather a test of the relationship between the AB5C and NEO to all the other measures.

The second point also needs further clarification. Factor analysis does not necessarily produce scales that are maximally distinctive (as stated) — this is driven by analytic decisions made in the factoring process. Many hierarchical scales have highly correlated sub factors, either by design or due to the nature of the domain being measured. For the BFAS, the distinctiveness of the scales was driven by the use of orthogonal rotations in the second stage of scale development. There is benefit to knowing the orientation of the two BFAS aspects relative to each of the 5 factors *excluding* the NEO and AB5C data but this would require new network plots. The current plots are misleading (with respect to this point specifically) because they are heavily influenced by the same data originally used to derive the BFAS.

This leaves only the 3rd “test”, which does not provide very compelling support for the BFAS. I don’t think that makes it less interesting; in fact, the authors may benefit from noting that I have had considerable difficulty replicating the BFAS structure in independent samples. But, having only this one point changes the tone of that whole section as the section currently seems to be supportive of the BFAS model.

We now provide a better introduction to this section. On p. 9 of the introduction, we now write, 

“[W]e showcase how this facet atlas can be used to better understand particular constructs by examining facets measuring impulsivity, and we illustrate how it can test structural hypotheses contained within measures by examining the Big Five Aspect Scales.”

Regarding the first test, it is not true that BFAS scales must be in the core of each domain because they were derived from IPIP versions of AB5C and NEO items. Indeed, as a consequence of the orthogonal factor analytic process used to derive the BFAS, which maximized the distinctiveness of the two aspect scales within each domain, it is possible that the two scales “split the middle” and do not measure the core. We believe that this test is therefore informative.

Regarding the second test, we do not mean to imply that all factor analyses create maximally distinctive scales. On p. 18, we now write, 

“A consequence of the specific factor-analytic procedure used to derive the BFAS is that, because the two aspect scales in each domain were rotated to be orthogonal, they should each measure distinctly different content.”

***

Reviewer #4: 

1. I’ll note my identity at the outset as Doug Samuel as this is relevant in terms of possible Conflicts of Interest. Primarily in that I have collaborated with some of these authors recently. I have also worked with Dr. Hopwood on numerous projects in the past and consider him a friend. I also note that my graduate student (Meredith Bucher) and I have been doing somewhat related work in terms of trying to organize facet scales. We have done so using the AB5C framework explicitly (Bucher & Samuel, 2018 in Journal of Personality Assessment; and other Bucher & Samuel that was just accepted pending revisions at Journal of Research in Personality). I don’t see either of our papers as competing with the present effort, but do want to be upfront about an possible COIs as I suggest references below.

This is a paper that uses the Eugene Springfield Community Sample (ESCS) to conduct a network analysis of 268 lower-order scales, across 13 personality measures to determine how closely the relate to each other and arrive at an estimate of which content was most common across the different measures. This led to an interesting product of a web-based app that would allow this information to be adapted and used by researchers or clinicians. I found this paper to be clear and easy to read. It had a number of older citations that showed a good command of seminal literature. That said, the authors unfortunately missed some very relevant citations that are key to this paper, including a major one that undertook very similar analyses in this same dataset. Woods and Anderson (2016) also used the ESCS to examine the degree of overlap of facet scales with the ultimate goal of creating a “periodic table of personality” (Woods, S. A., & Anderson, N. R. (2016). Toward a periodic table of personality: Mapping personality scales between the Five-Factor Model and the circumplex model. Journal of Applied Psychology, 101, 582–604.). This Woods and Anderson paper covers much of the same ground as the present paper and as such it simply must be incorporated into the background. My opinion is that there is sufficient differences in method, as well as product that it does not suggest this current paper shouldn’t be published, but it will need to be contextualized in terms of what has already been done and compared to those results. I think this becomes most tricky in terms of the wording. That paper aimed to provide a “periodic table of personality” and this one aims for a “personality atlas.” IMO, there may be differences there, but they are subtle and the present authors will need to take great care to explain how this is different and/or the same.

We thank the reviewer for pointing out this citation. This paper serves as a nice complement to the present investigation of blended facets, and as such we summarize it on p. 5.

“One past effort sought to create a circumplicial periodic table of blended facets, using 8 questionnaires assessed in the same dataset as the present study and two questionnaires assessed in a different sample [30]. Woods and Anderson (2016) identified 22 common blends and provided a term for each (for example, high conscientiousness and low extraversion was termed cautiousness). Notably, although they found content to represent most Big Five blends, they found very little content measuring either positive blends or contrasts of agreeableness and conscientiousness. In the present study, we investigate if this finding holds when additional facets are included from other hierarchical personality inventories, and we investigate whether some facets contain blends of three or more Big Five domains.”

We also cite this paper in the discussion section, on p. 20.

Although both studies provide analyses facet blends using primarily ESCS data, we believe that the two papers have quite different (and complimentary) goals. Specifically, the periodic table of personality aggregates different facet scales from many inventories, creating a summary of each blend of two Big Five traits within a circumplex, whereas this facet atlas retains the focus on each individual facet scale and uses a hierarchical approach to personality structure.

2. Additionally, the authors should consider citing a 2016 editorial by Ziegler & Backstrom that appeared in European Journal of Psychological Assessment (issue 32, pages 105-110) that also took a look at trying to sort through various facets. Another work to consider that is aimed at a similar goal is by David Condon and Bill Revelle. I am honestly not sure what is the best citation for it, but this one might be it (https://openpsychologydata.metajnl.com/articles/10.5334/jopd.32/). My understanding was that they were aiming to create an omnibus measure that represented a consensus list of facets across many inventories.

We have added citations of Ziegler and Backstrom (2016) on p. 3, and Condon et al. (2017) on p. 23.

3. As I mentioned, Meredith Bucher and I have published one paper that created a short form of the AB5C and then used new data to sort major faceted inventories (HEXACO and NEO) into the that context. The AB5C specifically is very relevant to the present paper as the authors repeatedly focus on the circumplex tradition and the promise it holds for understanding facets (page 4 of the present paper)…which is of course the entire idea behind the AB5C, so it would be helpful to cite as support here. In our 2018 paper (JPA; issue 101:1, 16-24) we found that some domain blends were less well-represented than others, so this is quite relevant to current discussion. In particular, our 2nd paper is specifically looked at those facets of the AB5C that weren’t performing quite as expected in terms of primary and secondary loadings to see if new items (from the IPIP) would improve this. Interestingly, we find that the specific facet that blends high agreeableness and low conscientiousness was not able to be measured well. We interpret this as being an area of personality space that may well be less populated. We don’t go so far as to say its vacant, but the combination of findings across studies suggest there are few, if any, words in the lexicon that map into this space and we couldn’t not locate suitable items to measure that combination. In short, it seems to be an area of the personality space that seems to have been uncommon (or unimportant) that it was not encoded into language or common measures. We hope to have this accepted soon, but I’d be happy to share the version informally. As it pertains here, I’d appreciate the authors digging more into those less populated facets in their writing.

 We were also quite intrigued by this finding, and we now discuss it further on pp. 20-21.

“Two particular types of facet blends, between extraversion and openness, and between agreeableness and conscientiousness, were uncommon in the facet atlas. This finding was relatively surprising, as these two pairs of domains are typically intercorrelated [65]. The paucity of blends between facets of extraversion and openness does not appear to reflect empty space, as we did identify a few positive blends (TCI exploratory excitability and AB5C leadership) and a few contrasting blends (AB5C introspection and sociability). Also, past circumplicial research has also found that facets measuring ingenuity, creativity, and bold leadership measure a blend of high openness and high extraversion [30,67]. Rather, it seems that scales measuring blends of extraversion and openness are just uncommon. Developing scales that explicitly measure a blend of these traits may be useful, as they form the metatrait of plasticity and have been theorized to function in tandem as part of the approach system (68). In comparison, the lack of blended content between agreeableness and conscientiousness may reflect trait space that is necessarily sparser. We found one negative blend (AB5C rationality) and three positive blends (AB5C dutifulness, AB5C morality, and HEXACO fairness), and other research has similarly struggled to identify blended content between these two traits (69). We note that each of these four facets connotes some sort of interpersonally-focused rule adherence, which is a common behavior in everyday life but does not seem to be well-encoded into the language of stable individual differences (as evidenced by the fact that we cannot identify a single-adjective term to describe this kind of behavior). Future research may wish to further delve into the personality space occupied by a blend of agreeableness and conscientiousness, and to develop scales that explicitly measure this content. Doing so could augur a more comprehensive approach to personality assessment.”

4. The authors do a reasonable job of noting the strengths and weaknesses of the ESCS. The demographics are less than ideal, but it has a great set of measures (that is thorough, but not comprehensive). That said, more needed to be said about the complications of correlating measures that were administered up to a decade apart from each other. This is important as it relies upon the input to dictate the output. These are more “consensus” than completeness (e.g., it won’t “resolve” anything”). A key limitation is that the “items in some scales were administered over a period of multiple years” as noted on page 9. Some scales you are correlating are measured a decade apart. What impact might this have? 

In initial drafts of this article, we statistically corrected for measurement unreliability according to the formulas for decay of personality stability over time derived in Anusic and Schimmack (2016). Interestingly, because the decay reaches its asymptote rather quickly, according to their meta-analytically derived formula (after around 3 years), these corrections had very little impact on the correlational structure of the data. Nonetheless, we agree that this is worth noting in the paper. On p. 12, we now write, 

“Because some ESCS scales were administered years apart from one another, some correlations may be somewhat attenuated (see (60)).”

5. It was also not clear how big the sample was for the network analyses. It is not clear, but I would understand based on the variability in Table 3 that some sort of deletion was used for given correlations. But when this was for the entire collection of measures, did you also use listwise deletion? If so what was the final sample size? Or were missing data imputed?

On p. 11 of the methods section, we now clarify that

“For each Big Five domain, we then estimated a full (rather than partial) correlation matrix, using pairwise deletion, which described associations among facets.” 

In the abstract and methods section, we now clarify that pairwise Ns = 571-948.

6. I also encourage the authors to clearly define what they mean by “core.” Given the method, it means here that is the most common within the measures used here. It is NOT the lexical core or mean to be the “center” of the domain. I personally would prefer a term like “consensus” or the “common denominator” as this more clearly reflects what it means. This has important implications for the discussion in particular as the authors wrote “Network theory suggests that change to a network’s central nodes may cascade through the rest of the network [32]. This idea is appealing to intervention researchers who seek to use focal interventions to enact relatively far-reaching personality change [66]. Figures 1-5 provide some guidance about which facets are more or less core, and thus may represent better or worse targets for domain wide change efforts.” IMO, this is not necessarily correct. These analyses tell us what is most commonly included among these 13 measures. That doesn’t mean they are at the center of the lexical factor, nor that it is causally core in the way that is implied by the above example. We would need more data to make that comparison; ideally longitudinal data.

Similarly, what you call periphery doesn’t mean it is “less relevant” (although it might), but here it only means it is less common across these measures. For all we know, a very uncommonly assessed facet might turn out to the be the most “central” or “core” of a domain.

On p. 5, we define core in empirical terms,

“Some facets are situated conceptually and empirically at a domain’s core, as indicated by strong correlations with many other facets within that domain.”

Because the content of personality questionnaires reflects the test constructors’ theories about personality structure, the least commonly assessed facets are those least often theorized to be important to measuring domain. Nevertheless, if these uncommonly assessed facets correlate highly with other facets, they will appear near a domain’s core. In this way, patterns of covariance and frequency of inclusion work together to allow us to accurately map a domain’s core and periphery from a transtheoretical perspective. We reiterate this tests-as-theories logic to readers on p. 6:

“The abundance of personality measures in the ESCS provides a unique opportunity to identify a transtheoretical core to each Big Five domain by triangulating across personality facets from many inventories.”

This comment brings up a broader point, which is that structural theories based on patterns of covariance do not necessarily map onto reality. As such, if the term core is taken to mean something beyond our definition like “the common biological aspects underpinning a group of traits,” it is quite possible that an uncommon facet may turn out to be core. However, this perspective on personality structure is outside the scope of the present paper.

To address the second part of your comment, we acknowledge that whether change cascades from the core is rather speculative and best tested using longitudinal data. As such, we have removed that paragraph from the manuscript. 

Minor points:

1. I was not certain of the point of the BFAS-specific focus in the discussion. Why single out the BFAS over any of the other measures?

We use the BFAS to illustrate how hypotheses about the structural properties of an instrument can be tested using network visualization. We now clarify this on p. 9 of the introduction, as well as in the abstract. In particular, the BFAS serves as a good illustrative example for this paper because it contains relatively few scales (and can thus be illustrated relatively briefly) and because the logic and hypotheses underlying its construction is laid out clearly in DeYoung et al. (2007).

2. I was surprised to see the authors correct using Cronbach’s alpha. Of course, as the authors know, alpha is not an index of true reliability. Les Morey’s address at SPA in San Francisco a few years ago provided a number of examples of how alpha was not a measure of reliability and why correcting using that could lead to skewed estimates of this. Test-retest dependability would be a much stronger correction. In the absence of test-retest over a short period (Watson, 2004), I would encourage the authors to just abandon corrections at all. I realize that may sound extreme so I’d also be content if they simply provided the uncorrected results as well.

We agree that these corrections are rather rough, as we note in the limitations section. However, as we write on p. 12, 

“Because the facet scales in this study varied in length from 1 to 46 items, it was especially important to correct for measurement error due to scale length … we lacked item-level information for some scales.”

Unfortunately, this lack of item-level data (and test-retest data) means that our only option to correct for unreliability was by using alpha. This correction for unreliability can be toggled in the facet galaxy app, allowing readers to compare corrected and uncorrected results side-by-side. When this is done, the major effect on the networks is that brief facet scales (such as those from the BFI) are pushed towards the network peripheries because of their lower reliability. 

3. On page ?? you say with regard to blended traits that “basic and applied personality researchers seem reluctant to incorporate this complex, blended reality into personality assessment and theory.” Can you give an example of the relucatance? I don’t personally see such a reluctance, but maybe I’m not clear on what is being communicated.

On p. 19, we now clarify, 

“[B]asic and applied personality researchers seem reluctant to incorporate this complex, blended reality into personality assessment and theory, as evidenced by the simple structure implied in most non-circumplicial personality measures (e.g. [8,10,12]).”

---

## [Editor Report · Decision Letter 1]

16 Jul 2020

A facet atlas: visualizing networks that describe the blends, cores, and peripheries of personality structure

PONE-D-20-09874R1

Dear Dr. Schwaba,

We’re pleased to inform you that your manuscript has been judged scientifically suitable for publication and will be formally accepted for publication once it meets all outstanding technical requirements.

Kind regards,

Frantisek Sudzina

Academic Editor

PLOS ONE

---

## [Editor Report · Acceptance letter]

20 Jul 2020

PONE-D-20-09874R1 

A facet atlas: visualizing networks that describe the blends, cores, and peripheries of personality structure 

Dear Dr. Schwaba:

I'm pleased to inform you that your manuscript has been deemed suitable for publication in PLOS ONE. Congratulations! Your manuscript is now with our production department. 

Kind regards, 

on behalf of

Dr. Frantisek Sudzina 

Academic Editor

PLOS ONE